# Sensing of cytosolic LPS through caspy2 pyrin domain mediates noncanonical inflammasome activation in zebrafish

Dahai Yang[1], Xin Zheng[1], Shouwen Chen[1], Zhuang Wang[1], Wenting Xu[1], Jinchao Tan[1], Tianjian Hu[1], Mingyu Hou[1], Wenhui Wang[1], Zhaoyan Gu[1], Qiyao Wang[1,2,3,4], Ruilin Zhang[4,5], Yuanxing Zhang[1,2,3,4] & Qin Liu[1,2,3,4]

The noncanonical inflammasome is critical for cytosolic sensing of Gram-negative pathogens. Here, we show that bacterial infection induces caspy2 activation in zebrafish fibroblasts, which mediates pyroptosis via a caspase-5-like activity. Zebrafish caspy2 binds directly to lipopolysaccharide via the N-terminal pyrin death domain, resulting in caspy2 oligomerization, which is critical for pyroptosis. Furthermore, we show that caspy2 is highly expressed in the zebrafish gut and is activated during infection. Knockdown of *caspy2* expression impairs the ability of zebrafish to restrict bacterial invasion in vivo, and protects larvae from lethal sepsis. Collectively, our results identify a crucial event in the evolution of pattern recognition into the death domain superfamily-mediated intracellular lipopolysaccharide-sensing pathway in innate immunity.

[1] State Key Laboratory of Bioreactor Engineering, East China University of Science and Technology, Shanghai 200237, China. [2] Laboratory for Marine Biology and Biotechnology, Qingdao National Laboratory for Marine Science and Technology, Qingdao 266071, China. [3] Shanghai Engineering Research Center of Marine Cultured Animal Vaccines, Shanghai 200237, China. [4] Shanghai Collaborative Innovation Center for Biomanufacturing, Shanghai 200237, China. [5] State Key Laboratory of Genetic Engineering, School of Life Sciences, Fudan University, Shanghai 200433, China. Correspondence and requests for materials should be addressed to Q.L. (email: qinliu@ecust.edu.cn)

The noncanonical inflammasome is critical for the cytosolic sensing of several Gram-negative pathogens, and its discovery might expand the spectrum of known pyroptosis mediators in mammals[1–5]. In humans, caspase-4 and caspase-5 seem to function similarly to mouse caspase-11, and are activated by intracellular lipopolysaccharide (LPS) to induce gasdermin D cleavage-triggered cell pyroptosis[5–7], which has proinflammatory effects and mediates innate immune defense against a variety of pathogens. Given the important function of the noncanonical inflammasome in mammalian immune systems, understanding related signaling pathway in lower vertebrates is critical.

The zebrafish is an important vertebrate organism to study host–pathogen interactions and immunity[8]. To date, at least five functional caspases have been identified in zebrafish, including the inflammatory caspy (also named as caspase-A), caspy2 (also named as caspase-B), and caspase-C[9–11]. The zebrafish genome contains caspase-C and a caspase-C-like protease, which have highest similarity to human caspase-4; however, these caspase variants do not have apoptotic protein domains in their N-terminal regions[9]. In addition, the catalytic domains of zebrafish caspy and caspy2 have highest homology with those of human caspase-1 and caspase-5, with sequence similarity of 54 and 57%, respectively[9–11]. Interestingly, both caspy and caspy2 have a pyrin domain at their N-terminal instead of the typical caspase activation and recruitment domain (CARD) of inflammatory caspases[9–11]. Compared with the identified role of caspy in mediating canonical inflammasome activation signaling[11–14], the biological function of caspy2 is unclear. Previous work has shown that caspy2 has the highest homology to human caspase-4/5, but cannot interact with apoptosis-associated speck-like containing a CARD (ASC)[10]. Enzyme activity analysis showed that caspy2 has a preferred caspase-5 substrate specificity[10]. Moreover, caspy2-transfected mammalian cells have morphological features of adherent cells undergoing apoptosis, such as rounding and membrane blebbing[10,15]. However, whether the zebrafish caspy2 binds cytosolic LPS and induces noncanonical inflammasome mediated pyroptosis in lower vertebrates is unclear.

Here, we report the functional identification and characterization of zebrafish caspy2, and provide evidence that its N-terminal pyrin death domain (PYD) is critical for LPS binding, and responsible for noncanonical inflammasome activation in zebrafish nonmyeloid cells. In addition, we show that caspy2 is highly expressed in the gut, and has an essential function in restricting bacterial infection in intestinal sites. Moreover, knockdown of caspy2 protects larvae from lethal sepsis. Our study first establishes the existence of caspy2-mediated noncanonical inflammasome activation in zebrafish, and clarifies the role of caspy2 activation in promoting host defense, which provides important insight into the evolution of pattern recognition in the death domain superfamily-mediated intracellular LPS-sensing pathway for innate immunity.

## Results

### Caspase-5 activity-mediated pyroptosis in zebrafish fibroblasts.

To analyze pyroptosis in zebrafish cells, we infected zebrafish fibroblasts (ZF4) with 0909I *Edwardsiella piscicida*, a hemolysin-overexpressing strain that specifically promotes caspase-4- and gasdermin D-gated noncanonical inflammasome activation-dependent cell death in mammalian HeLa cells (Supplementary Fig. 1). ZF4 cells exhibited pyroptotic morphological features, such as cell swelling, membrane rupture (as shown by propidium iodide (PI)-positive staining, Supplementary Fig. 2), and significant cytotoxicity (Fig. 1a) following infection with 0909I *E. piscicida*. The morphology of ZF4 cells infected with wild-type *E.*

*piscicida* (EIB202) was similar to that of untreated cells (Supplementary Fig. 2).

To investigate the link between pyroptosis and caspase activity in zebrafish fibroblasts, we assessed the ability of cell extracts to cleave a panel of fluorogenic and chromogenic peptide substrates of mammalian caspases. Notably, extracts from 0909I *E. piscicida*-infected ZF4 cells preferentially cleaved the fluorogenic substrate of human caspase-5, Ac-Trp-Glu-His-Asp-7-amino-4-methyl-coumarin (Ac-WEHD-AFC), but did not cleave the other caspase substrates tested (Fig. 1b). The cytotoxicity and pyroptotic morphology induced by 0909I *E. piscicida* were effectively inhibited by Z-WEHD-FMK and Z-VAD-FMK, which are caspase-5 and pan-caspase inhibitors, respectively (Fig. 1c, d). In contrast, Z-YVAD-FMK, Ac-DEVD-CHO, and Ac-LEVD-CHO, which inhibit caspase-1, caspase-3/7, and caspase-4, respectively, did not restrict the cytotoxicity or pyroptotic morphological changes in 0909I *E. piscicida*-infected ZF4 cells (Fig. 1c, d). Thus, these results indicated that the induction of pyroptotic morphology by 0909I *E. piscicida* in zebrafish fibroblasts is dependent on a caspase-5-like activity.

To further confirm the noncanonical inflammasome activation exists in zebrafish fibroblasts, we primed ZF4 cells with Pam3CSK4 and then stimulated them with cholera toxin B subunit (CTB) plus LPS, a commonly used stimulus that activates the noncanonical inflammasome. Consistent with 0909I *E. piscicida* infection, ZF4 cells exhibited significant cytotoxicity (Fig. 1e). The cytotoxicity induced by cytosolic LPS delivery was effectively inhibited by Z-WEHD-FMK and Z-VAD-FMK, but not Z-YVAD-FMK, Ac-DEVD-CHO, and Ac-LEVD-CHO (Fig. 1f). Moreover, lysates of LPS-transfected ZF4 cells showed significant preferential cleavage of Ac-WEHD-AFC, but did not cleave the other caspase substrates tested (Fig. 1g). Taken together, these results suggested that the noncanonical inflammasome is present in zebrafish.

According to previous research, both caspy and caspy2 are active caspases with different substrate specificity. Caspy preferentially cleaved Ac-YVAD-AMC, a caspase-1 substrate, whereas caspy2 was more active on Ac-WEHD-AFC, a preferred substrate of caspase-5[10,11]. Based on our results above, the pyroptosis induced in ZF4 cells is dependent on caspase-5-like activity, thus, we expect that caspy2 is an enzymatically active caspase in zebrafish, which might responsible for noncanonical inflammasome activation. As a more direct test of caspy2 activity, a panel of fluorogenic and chromogenic peptide substrates of mammalian caspases were incubated with caspy2 immunoprecipitated from extracts of cells transfected with a caspy2-expressing plasmid. Notably, caspy2 activity was highest with Ac-WEHD-AFC, a preferred substrate of caspase-5, and it did not cleave the other substrates (Supplementary Fig. 3a). Determination of the substrate-binding specificity of caspy2 to Ac-WEHD-AFC showed that this enzyme exhibited strong substrate concentration-dependent activity (Supplementary Fig. 3b). Furthermore, two caspy2 mutants, one carrying a Ser substitution of the critical Cys residue at position 296 (C296A, indicated in Supplementary Fig. 4) and another with an N-terminal pyrin truncation (ΔPYD, indicated in Supplementary Fig. 4), demonstrated impaired substrate-binding activity (Supplementary Fig. 3b). These resuslts suggested that zebrafish caspy2 is an enzymatically active caspase with caspase-5-like substrate specificity, which might play an important role in inducing pyroptosis in zebrafish fibroblasts.

### Caspy2 activation is critical for bacterial infection-induced pyroptosis.

To further validate the role of caspy2 in 0909I *E. piscicida* infection-induced pyroptosis in zebrafish fibroblasts, the

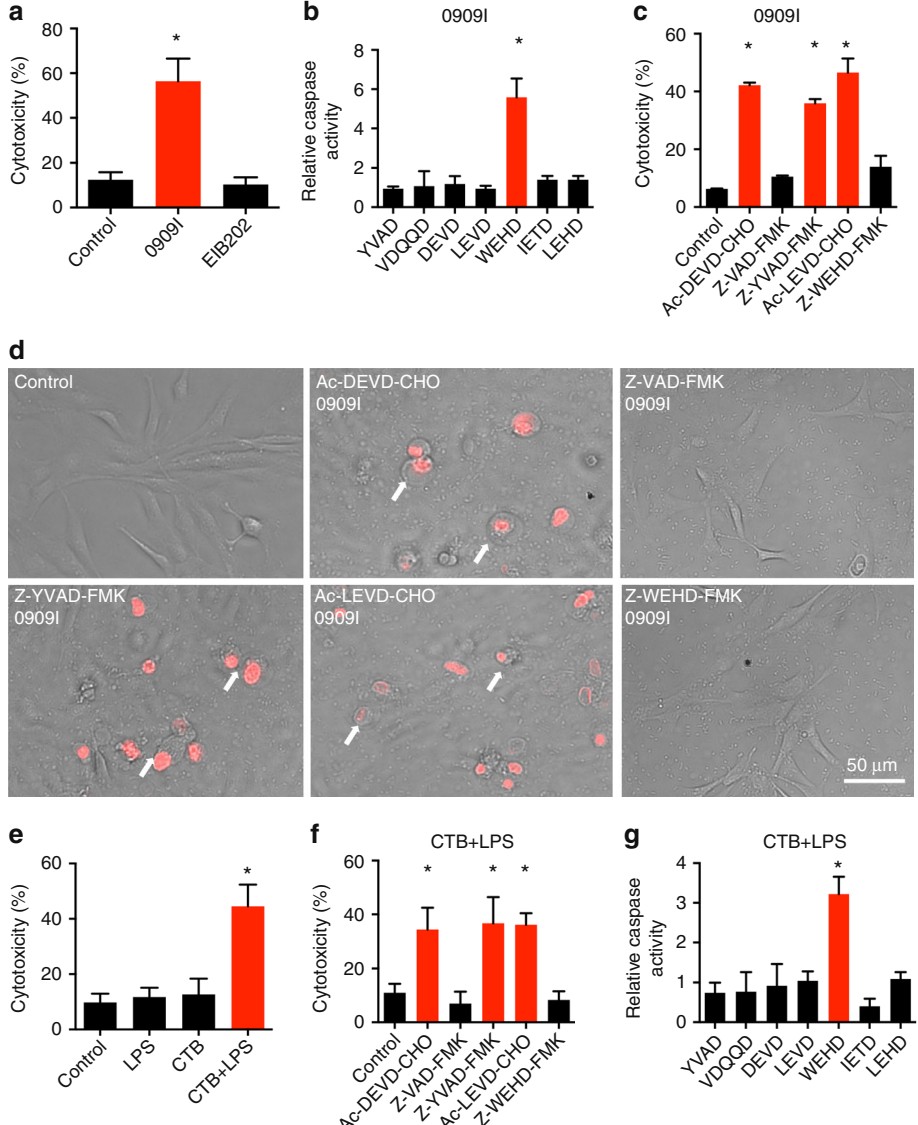

**Fig. 1** Caspase-5-like activity is essential for pyroptosis in zebrafish fibroblasts. **a** ZF4 zebrafish fibroblasts were infected with wild-type (EIB202) or 0909I *E. piscicida* for 2 h at a multiplicity of infection (MOI) of 50, or left uninfected. Supernatants from the indicated ZF4 cells were analyzed for cell death, as measured by lactate dehydrogenase (LDH) release. **b** ZF4 cells were infected with 0909I *E. piscicida* for 2 h at an MOI of 50. Relative caspase activity was then measured by incubating cell lysates with fluorogenic and chromogenic substrates of caspase-1 (YVAD), caspase-2 (VDQQD), caspase-3/7 (DEVD), caspase-4 (LEVD), caspase-5 (WEHD), caspase-8 (IETD), and caspase-9 (LEHD). **c, d** ZF4 cells were treated with caspase-3/7, pan-caspase, caspase-1, caspase-4, and caspase-5 inhibitors (Ac-DEVD-CHO, Z-VAD-FMK, Z-YVAD-FMK, Ac-LEVD-CHO, and Z-WEHD-FMK, respectively). **c** LDH release for cell death was measured 2 h after 0909I *E. piscicida* infection. **d** Images were taken after 0909I *E. piscicida* infection for 2 h. Propidium iodide (PI) was added to detect the loss of plasma membrane integrity. Arrows indicate cells exhibiting pyroptotic-like features. Scale bar, 50 μm. **e–g** ZF4 cells were primed with Pam3CSK4 for 4 h, before being stimulated with cholera toxin B subunit (CTB) plus LPS, LPS, or CTB alone for 12 h. **e** Supernatants from the indicated ZF4 cells were analyzed for cell death, as measured by lactate dehydrogenase (LDH) release. **f** Cytosolic LPS-delivered ZF4 cells were treated with indicated caspase inhibitors as in Fig. 1c, Supernatants from the indicated ZF4 cells were analyzed for cell death, as measured by lactate dehydrogenase (LDH) release. **g** Relative caspases activity was measured by incubating cell lysates with indicated fluorogenic and chromogenic substrates as in Fig. 1b. **a–g** Results are representative of at least three independent experiments, and error bars denote the SD of triplicate wells. *$p < 0.05$ (*t* test)

CRISPR/Cas9 genome-editing tool was applied to knockout (KO) *caspy2* in ZF4 cells, using a *caspy2*-specific guide RNA (gRNA) (Supplementary Fig. 5). The pSpCas9(BB)-*caspy2* gRNA-green flourescent plasmid (GFP) plasmids were used at 1 μg/ml and resulted in disruption of the *caspy2* locus of transfected ZF4 cells (Supplementary Fig. 5). Importantly, caspy2 deficiency impaired the development of the pyroptotic morphology induced by 0909I *E. piscicida* infection (Fig. 2a), and prevented the associated cytotoxicity (Fig. 2b). Moreover, when we infected

wild-type and caspy2-KO ZF4 cells with wild-type (EIB202) or 0909I *E. piscicida*, we observed a comparable transcriptional induction of caspy2 (Fig. 2c); however, the cleavage of pro-caspy2, as determined by immunoblotting using an antibody that recognizes p20, the cleaved form of caspy2, was only detected during 0909I *E. piscicida* infection (Fig. 2c). Taken together, these findings indicated that the existence of pyroptosis in zebrafish is mediated by caspy2 activation during 0909I *E. piscicida* infection.

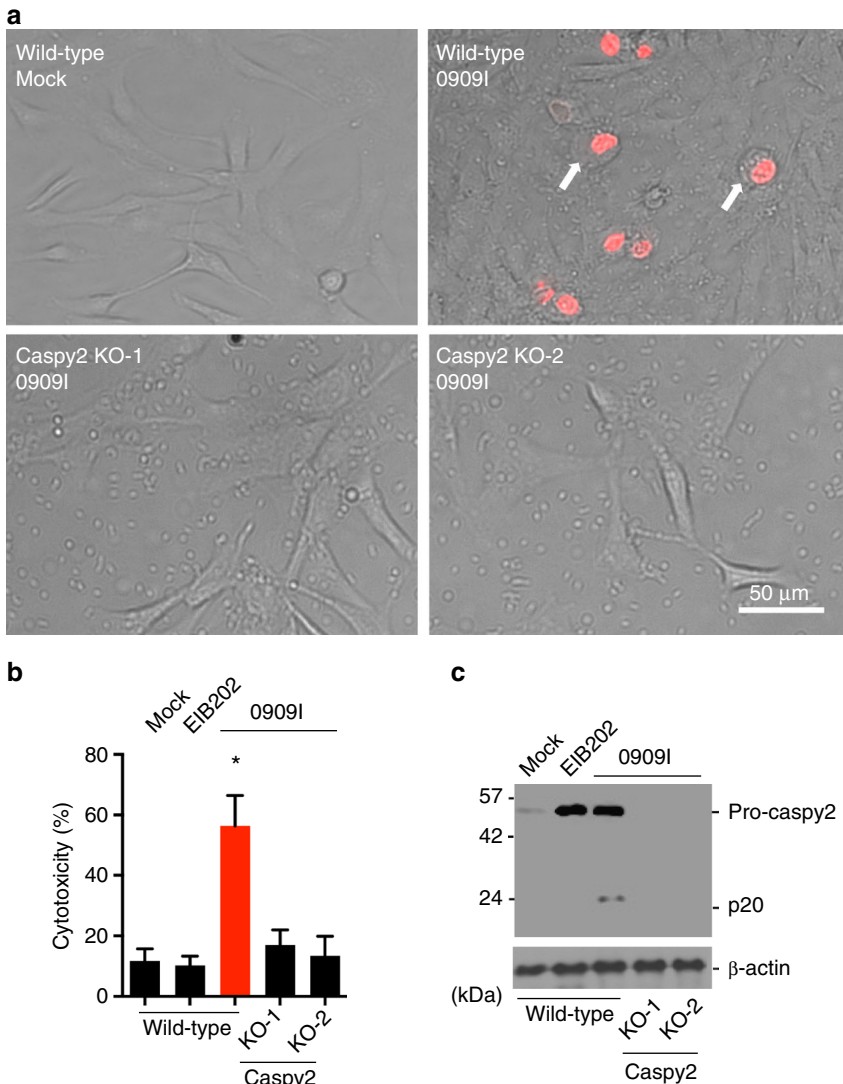

**Fig. 2** Caspy2 plays a critical role in pyroptosis in zebrafish fibroblasts. **a–c** Wild-type and *caspy2*-knockout (KO) ZF4 cells were infected with EIB202 or 0909I *E. piscicida* for 2 h at an multiplicity of infection (MOI) of 50, or left uninfected. **a** Images were taken as in Fig. 1d. Propidium iodide (PI) was added to detect the loss of plasma membrane integrity. Arrows signify cells with pyroptotic-like features. Scale bar, 50 μm. **b** Supernatants from the indicated ZF4 cells were analyzed for cell death measured by lactate dehydrogenase (LDH) release. **c** Mixtures of cell lysates and supernatants were subjected to immunoblotting. (**a–c**) Results are representative of at least three independent experiments, and error bars denote the SD of triplicate wells. $^{*}p < 0.05$ (*t* test)

**Caspy2 binds directly to LPS and undergoes oligomerization**. The N-terminal death domain superfamily (caspase-recruitment domain, CARD) of caspase-4/5 and caspase-11 can bind LPS and activate the noncanonical inflammasome[6]. However, the N-terminal domain of caspy2 is most homologous to that of human Cryopyrin/PYPAF1 (46% similarity), a pyrin domain containing Nod-family protein[10]. Sequence analysis also showed that the pyrin domain of caspy2 exhibited significant homology not only to pyrin domains, but also to the CARDs of Xenopus caspase-1 (49% similarity) and bovine caspase-13 (45% similarity, previously known as human caspase-13)[10]. Thus, we speculated that the pyrin domain superfamily might possess a similar function in directly binding LPS. To test this idea, we carried out a series of pulldown assays to test the LPS-binding affinity and specificity of caspy2 and caspy. Biotinylated-ultrapure LPS efficiently precipitated caspy2, caspy2 (C296A) and caspy2 PYD, but not caspy2 (ΔPYD), caspy or caspy PYD (Fig. 3a). However, neither caspy nor caspy2 could be pulled down by biotinylated-Pam3CSK4, another immunostimulatory peptide from bacterial

lipoprotein (Fig. 3a). In addition, in enzyme linked immunosorbent assays (ELISAs) of LPS-binding activity. Similar to caspase-11, caspy2, caspy2 (C296A), and caspy2 PYD showed strong affinity to immobilized LPS, with the number of LPS-caspy2 complexes increasing in a concentration-dependent manner (Supplementary Fig. 6). However, caspy2 (ΔPYD), caspy, caspy PYD, and caspase-1 displayed no affinity for LPS (Supplementary Fig. 6), demonstrating that the caspy2 N-terminal pyrin domain is crucial for LPS recognition in zebrafish (Supplementary Fig. 6). Consistently, neither caspy nor caspy2 were not precipitated by biotinylated-Pam3CSK4 (Fig. 3a).

Recent studies have shown that caspase-4/5 and caspase-11 undergoes oligomerization upon LPS binding, which is required for the caspases' catalytic activity[6]. Consequently, we used 0.5% Triton X-100 to reduce the LPS-induced caspases into a smaller homogenous form on blue native polyacrylamide gel electrophoresis (PAGE)[6], followed by western blotting with anti-HA antibodies. We found that purified HA-tagged caspy2 and caspy2 PYD from transfected HEK293T cell were oligomerized upon

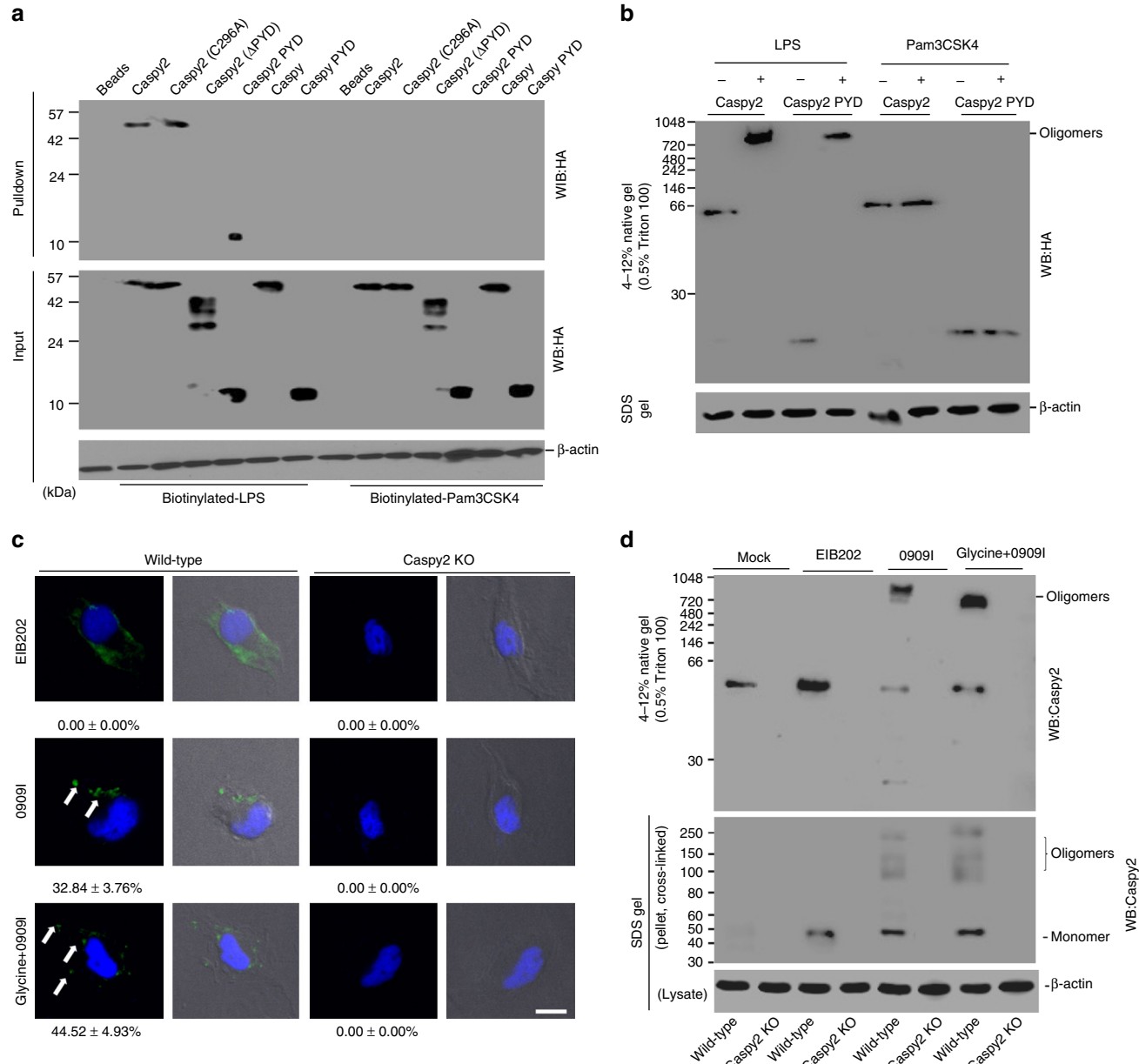

**Fig. 3** Caspy2 directly binds to LPS to induce its oligomerization. **a** Streptavidin pulldown assays of the binding of biotin-conjugated LPS and Pam3CSK4 to purified HA-tagged caspy2, caspy2 (C296A), caspy2 (ΔPYD), caspy2 PYD, caspy and caspy PYD in transfected HEK293T cells. Shown are anti-HA immunoblots of pulled down proteins and total lysates (input). **b** Purified HA-tagged caspy2 and caspy2 PYD in transfected HEK293T cells were incubated with LPS or Pam3CSK4. The samples were analyzed by the pore-limited native gel electrophoresis. **c** Wild-type and *caspy2*-KO ZF4 cells were infected with wild-type (EIB202) or 0909I *E. piscicida* (with/without 10 mM glycine) for 2 h at a multiplicity of infection (MOI) of 50. Confocal laser scanning microscopic analysis of nuclei (2-(4-amidinophenyl)-1H-indole-6-carboxamidine (DAPI), blue) and caspy2 foci (green, white arrowheads). Direct interference contrast (DIC)/phase images are shown. Scale bar, 10 μm. Statistics of the percentages of cells showing signals for caspy2 foci are listed below. (Approximately 200 cells were counted in each sample. Mean ± SD of triplicate samples.) **d** Wild-type and *caspy2*-KO ZF4 cells were infected with wild-type (EIB202) or 0909I *E. piscicida* (with/without 10 mM glycine) for 2 h at an MOI of 50, or left untreated (Mock). The samples were analyzed by the pore-limited native gel electrophoresis and immunoblotting (upper panel). Cell lysates and DSS cross-linked pellets from wild-type and *caspy2*-KO ZF4 cells treated as indicated were analyzed by immunoblotting for caspy2 oligomerization (lower panel). WB western blot. Results are representative of at least three independent experiments

incubation with LPS, but not with Pam3CSK4 (Fig. 3b). Furthermore, to detect whether the caspy2 oligomerization occurs during bacterial infection, we utilized an antibody that recognizes the caspy2 N-terminal PYD to probe the caspy2 foci. A significantly higher quantity of caspy2 foci was observed in 0909I *E. piscicida*-infected wild-type ZF4 cells, compared with that in EIB202-infected cells (Fig. 3c), and no signaling was detected in *caspy2*-KO ZF4 cells infected with either bacterial strains (Fig. 3c).

Moreover, we also used the cytoprotectant glycine to inhibit pyroptotic cell lysis during 0909I *E. piscicida* infection, and observed more caspy2 foci in 0909I *E. piscicida*-infected wild-type ZF4 cells (Fig. 3c). Consistently, caspy2 treated with cross-linking agents remained as oligomers on the sodium dodecyl sulfate (SDS) gel, and oligomerization was triggered by infection with 0909I, but not with EIB202 *E. piscicida* (Fig. 3d). In addition, to assess caspy2 assembly during infection, wild-type or *caspy2*-KO

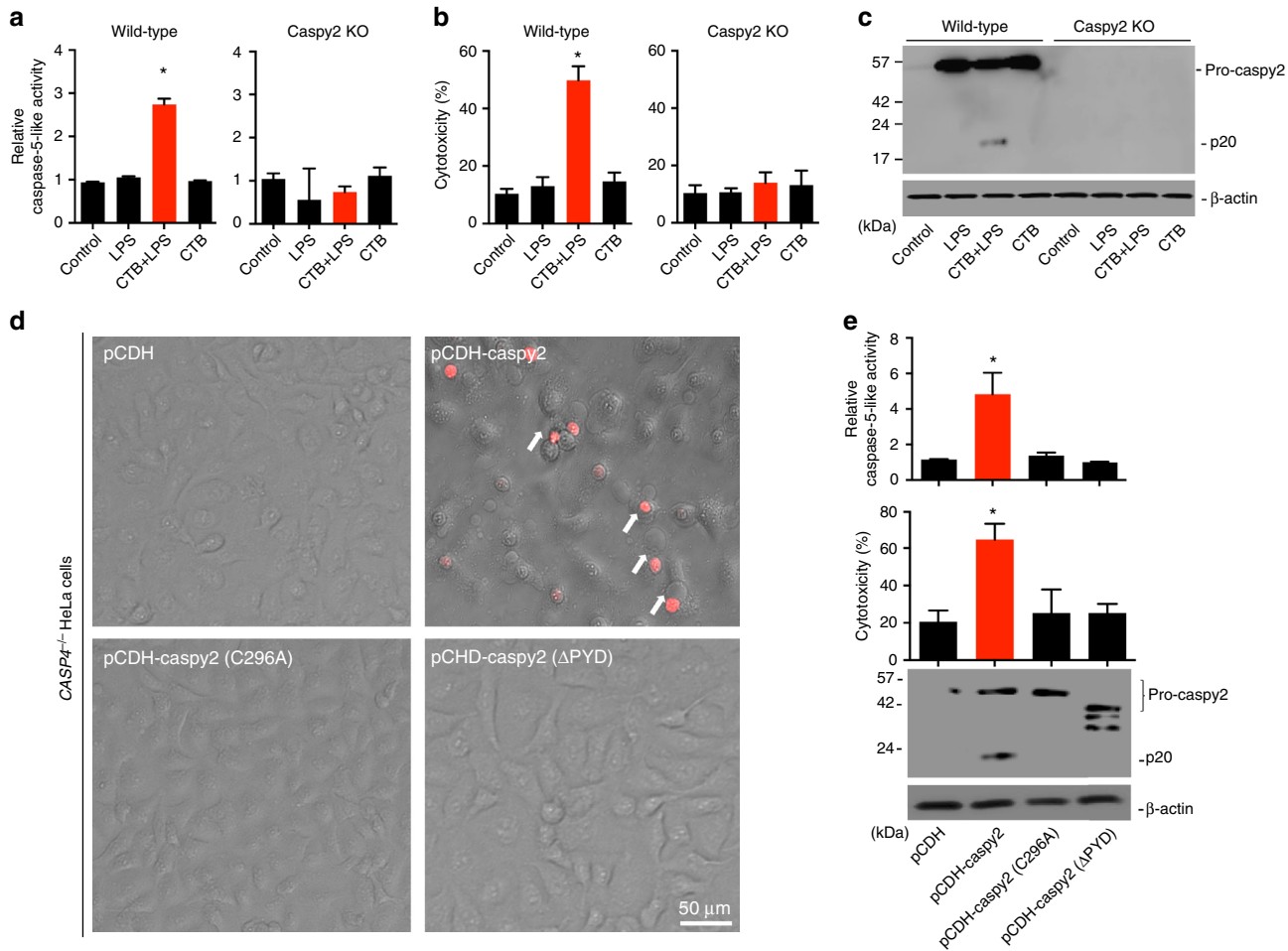

**Fig. 4** Intracellular LPS-triggered-caspy2 noncanonical inflammasome activation. **a-c** ZF4 cells were primed with Pam3CSK4 for 4 h, before being stimulated with cholera toxin B subunit (CTB) plus LPS, LPS, or CTB alone for 12 h. Relative caspase-5-like activity was measured by incubating cell lysates with fluorogenic and chromogenic caspase-5 substrates (WEHD) (**a**). Supernatants from the indicated HeLa cells were analyzed for cell death, as measured by lactate dehydrogenase (LDH) release (**b**). Mixtures of cell lysates and supernatants were subjected to immunoblotting (**c**). **d, e** *CASP4*$^{-/-}$ HeLa cells were transduced with a vector expressing wild-type caspy2, caspy2 (C296A), caspy2 (ΔPYD), or the empty vector. Cells were primed with LPS for 4 h, before being stimulated with CTB plus LPS for 12 h. Images were taken as in Fig. 1d (**d**). Propidium iodide (PI) was added to detect the loss of plasma membrane integrity. Arrows denote cells with pyroptotic-like features. Scale bar, 50 μm. Relative caspase-5-like activity and LDH release assays were conducted as in a and b (**e**). Immunoblotting for the caspy2 forms indicated is shown. **a–e** Results are representative of at least three independent experiments, and error bars denote the SD of triplicate wells. $^*p < 0.05$ (*t* test)

ZF4 cells were stimulated with 0909I or EIB202 *E. piscicida*, and digitonin-solublized cell lysates were resolved by blue native PAGE with 0.5% Triton X-100, followed by western blotting with anti-caspy2 antibodies. A large oligomeric complex containing caspy2 was induced in wild-type ZF4 cells upon infection with 0909I, but not with EIB202 *E. piscicida* (Fig. 3d). Taken together, these results suggested that zebrafish caspy2 directly binds LPS and forms oligomers responding to bacterial infection.

**Intracellular LPS stimulates caspy2 activation**. To dissect the cytoplasmic sensing of LPS by caspy2, we primed wild-type and *caspy2*-KO ZF4 cells with Pam3CSK4 and then stimulated them with CTB plus LPS, a commonly used stimulus that activates the noncanonical inflammasome. Lysates of wild-type, but not *caspy2*-KO, LPS-transfected ZF4 cells showed significant preferential cleavage of Ac-WEHD-AFC (Fig. 4a). Notably, caspy2 was important for intracellular LPS-induced cytotoxicity (Fig. 4b), and the formation of the p20 cleaved form of caspy2 was detected in wild-type LPS-transfected ZF4 cells (Fig. 4c). These results indicated that caspy2 is essential for intracellular

LPS-triggered noncanonical inflammasome activation in zebrafish.

To determine whether the N-terminal pyrin domain or the catalytic activity of caspy2 is required for pyroptosis, we expressed the wild-type, C296A, or ΔPYD form of this enzyme in *CASP4*$^{-/-}$ HeLa cells (Fig. 4d, e). Importantly, wild-type caspy2, but not the C296A or ΔPYD mutants, restored intracellular LPS-induced pyroptosis in these cells (Fig. 4d, e). In addition, we expressed the wild-type, C254S, or ΔCARD form of caspase-11 in *CASP4*$^{-/-}$ HeLa cells. Consistently, wild-type caspase-11, but not the C254S or ΔCARD mutants, restored intracellular LPS-induced pyroptosis in these cells (Supplementary Fig. 7). These observations suggested that the zebrafish caspy2 pyrin domain has a function similar to that of the caspase-11 CARD domain in recognizing intracellular LPS to mediate its oligomerization (Supplementary Fig. 8); in addition, the catalytic activity is required for pyroptosis in zebrafish.

**Caspy2 is highly expressed in developmental gut**. A previous study suggested that although *caspy2* is expressed during

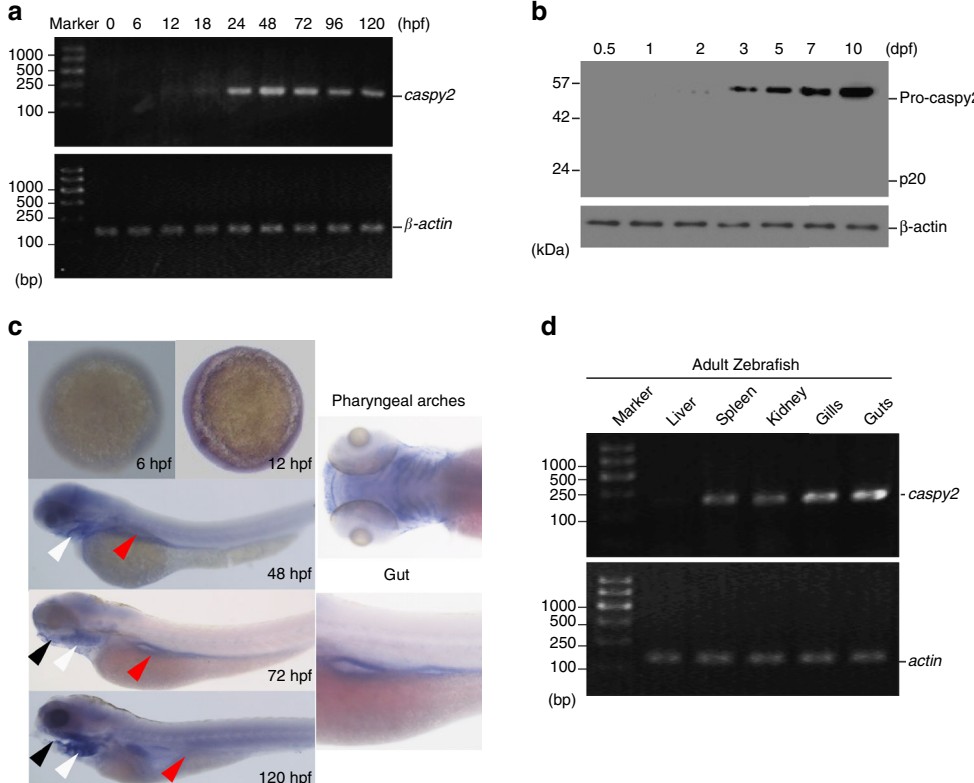

**Fig. 5** Pattern of *caspy2* expression in zebrafish. **a** Reverse transcription polymerase chain reaction (RT-PCR) analysis of relative *caspy2* expression at various stages of development. **b** Immunoblotting analysis of caspy2 levels at various stages of whole larvae development. **c** Whole-mount in situ hybridization targeting *caspy2* mRNA in zebrafish at the developmental stages indicated. Black arrowheads, mouth; white arrowheads, pharyngeal arches; red arrowheads, gut. **d** RT-PCR analysis of relative *caspy2* expression in the indicated adult zebrafish organs. **a–d** Results are representative of at least three independent experiments

zebrafish development, such expression is largely limited to the epidermis, mouth, and pharyngeal arches[10]. In the present study, we performed reverse transcription polymerase chain reaction (RT-PCR) to measure *caspy2* expression at various developmental stages. The results revealed that *caspy2* was expressed from 18 h postfertilization (hpf) (Fig. 5a). Consistent with this, expression of the caspy2 protein in the whole embryo was observed from 24 hpf, and was markedly induced at 5–7 days postfertilization (dpf) (Fig. 5b). To better understand the spatial pattern of *caspy2* expression throughout zebrafish development, we performed whole-mount in situ hybridization. Consistent with previous research[10], *caspy2* mRNA was detected in the mouth and pharyngeal arches from 48 hpf (black arrowheads and white arrowheads, respectively, in Fig. 5c). Interestingly, during this time, *caspy2* expression was clearly visible in the developing gut (red arrowheads in Fig. 5c). Moreover, in adult zebrafish, substantial *caspy2* mRNA expression was detected in the gills and gut (Fig. 5d), which was significantly stronger than that in the spleen, kidney, and liver. These data demonstrated that *caspy2* is preferentially expressed in zebrafish intestinal sites, suggesting that caspy2 might play a role in gut defenses.

In addition, we investigated the effect of caspy2 on zebrafish development. A morpholino oligonucleotide to block *caspy2* expression was designed and injected into the embryos at the one-cell stage (Supplementary Fig. 9a, b). Immunoblotting showed that this morpholino oligonucleotide (*caspy2*-MO) effectively knocked down *caspy2* up to and including 7 dpf (Supplementary Fig. 9c), and confirmed that the control-morpholino oligonucleotide (control-MO) containing five base mismatches had no effect on *caspy2* expression (Supplementary

Fig. 9c). The *caspy2* morphants displayed no phenotypic abnormalities, such as general tissue loss, cell death in the head, a curled body axis, and pericardial edema (Supplementary Fig. 9d), indicating that *caspy2* knockdown did not alter zebrafish embryonic development.

**Caspy2 is critical for gut inflammation and antibacterial defenses.** Given that the noncanonical inflammasome is critical in combating infection by enteric bacterial pathogens at the intestinal mucosal surface[3,16], we first assessed the expression of *caspy2* in zebrafish larvae at 120 hpf in response to bacterial infection by immersion (Supplementary Fig. 9a). The induction of *caspy2* expression and cleavage of pro-caspy2, as determined by immunoblotting of whole zebrafish larvae extracts using an antibody against p20, were detected during 0909I *E. piscicida* infection, but not with wild-type *E. piscicida* (EIB202) infection (Fig. 6a). These results demonstrate that caspy2 activation is significantly induced during bacterial infection in vivo, indicating its involvement in zebrafish immune defenses.

To analyze the in vivo role of caspy2 in promoting intestinal immune defenses, we performed a zebrafish larvae immersion infection procedure (Supplementary Fig. 9a). Following infection with $10^5$ colony forming units (CFU)/ml of 0909I *E. piscicida*, *caspy2*-MO larvae rapidly succumbed, whereas the control-MO larvae were unaffected (Fig. 6b). In agreement with a role for caspy2 in gut antibacterial defenses, *caspy2*-MO larvae had significantly higher gut pathogen loads (Fig. 6c, d). At early time points, initial gut colonization by 0909I *E. piscicida* appeared similar in *caspy2*-MO and control-MO larvae (Fig. 6c, d). However, by 24 hpi, it was dramatically enhanced in the former,

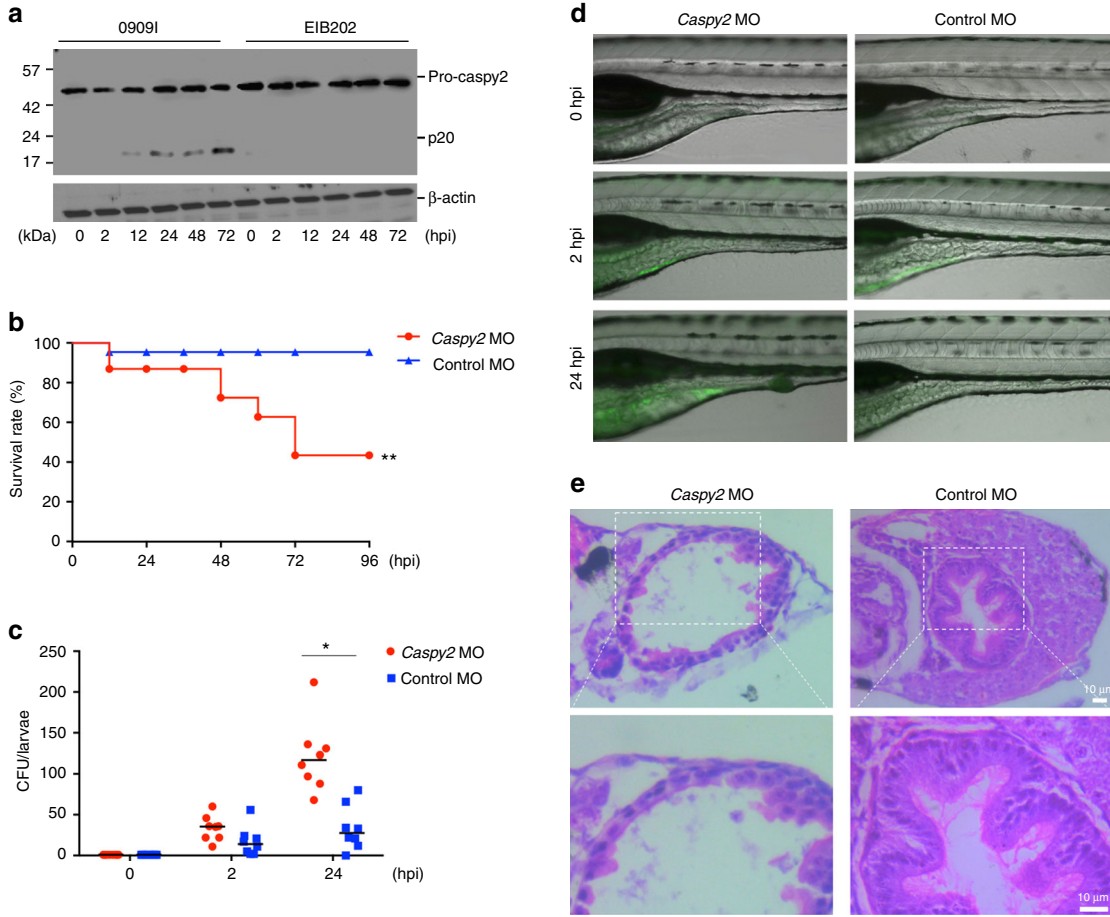

**Fig. 6** Caspy2 restricts bacterial colonization of the zebrafish gut in vivo. **a**–**e** Control-morpholino oligonucleotide (MO) or *caspy2*-MO zebrafish larvae at 5 days postfertilization (dpf) were infected by immersion with $10^5$ colony forming units (CFU)/ml of 0909I *E. piscicida*. **a** Zebrafish larvae (5 dpf) were infected by immersion with wild-type (EIB202) or 0909I *E. piscicida*. Immunoblotting analysis of caspy2 levels at the indicated post-infection time points. **b** Survival of zebrafish larvae was monitored for 4 days. ($n = 60$ for control-MO, $n = 60$ for *caspy2*-MO). Results are representative of at least three independent experiments, $^{**}p < 0.01$ (log-rank). **c** Zebrafish larvae were collected at the indicated postinfection time points, and homogenates were made for CFU counts. Each symbol represents the average counts of five larvae ($n = 100$ for control-MO, $n = 100$ for *caspy2*-MO). Results are representative of at least three independent experiments. $^*p < 0.05$ (ANOVA). **d** Images of infected zebrafish larvae. Green fluorescent protein (GFP)-0909I *E. piscicida*. **e** Representative images of hematoxylin and eosin (H&E) staining of gut sections from 0909I-infected control-MO or *caspy2*-MO zebrafish larvae. Square frame, intestinal wall morphology; Scale bar, 10 μm

but not the latter (Fig. 6c, d). Furthermore, unlike the control-MO larvae, *caspy2*-MO larvae exhibited prominent histopathological signs of gut inflammation after infection with 0909I *E. piscicida* (Fig. 6e). Thus, our results suggest that fish noncanonical inflammasome activation is important for bacterial clearance, playing a hitherto unrecognized role in the innate immunity of lower vertebrates.

**Caspy2 is required for lethal sepsis.** The predominant model for the experimental study of sepsis is the mouse, which is initiated by the administration of LPS or via pathogens infection[17–20]. Zebrafish embryos display many of the key pathophysiological features seen in human patients with sepsis, including dysregulated inflammatory responses (cytokine storm), tachycardia, and endothelial leakage through *E. coli* infection[20]. The noncanonical inflammasome processes was critical for septic shock in mammals[21,22]. In this study, we assessed the role of caspy2 in a zebrafish larval model of lethal sepsis[23] by administering a lethal dose of LPS by immersion to induce lethality in 5 dpf larvae. Control-MO larvae exposed by immersion to a lethal dose of LPS died rapidly, whereas the survival rate was higher among *caspy2*-

MO larvae given the same treatment (Supplementary Fig. 10a). Consistent with the morphology observed during 0909I *E. piscicida* infection, LPS-treated zebrafish displayed marked pericardial edema, and erosion of the tail fin and body axis in control-MO larvae after 24 hpi (Supplementary Fig. 10b). Moreover, the significant upregulation of IL-1β, TNF-α, IL-6, IL-8, IL-10, and IFN-γ in control-MO larvae was also observed at 12 hpi, compared with that in *caspy2*-MO larvae treated with LPS (Supplementary Fig. 10c). Taken together, these results indicated that activation of the noncanonical inflammasome in vivo by LPS causes rapid mortality via the noncanonical inflammasome pathway in zebrafish.

## Discussion

Most previous studies concerning the assembly of the noncanonical inflammasome during inflammatory responses have focused on the functions of caspase-4/5 or caspase-11 in mammalian cells[24,25]. In mouse macrophages, noncanonical inflammasomes are assembled by direct binding of LPS through the CARD domain of procaspase-11[6,26]. Caspase-4 and caspase-5 are the human counterparts of mouse caspase-11, which were

also found to activate the noncanonical inflammasome in human myeloid cells[21,22]. Similar to the noncanonical inflammasome assembly in mice, procaspase-4 and/or procaspase-5 in human cells interact directly with LPS, and this interaction is mediated by binding of the CARD domain[27–29]. In this study, we revealed that zebrafish caspy2 shows a caspase-5-like activity that is critical for noncanonical inflammasome activation in non-myeloid cells (Fig. 1). However, unlike the CARD domain as found at the N-terminal of caspase-4/5/11 in mammals, the N-terminal domain of caspy2 is most homologous to the pyrin domain containing NOD-family protein[10]. Interestingly, our data found that the pyrin domain of caspy2 is critical in mediating intracellular LPS binding, resulting in caspy2 oligomerization and activating the noncanonical inflammasome in zebrafish (Supplementary Fig. 8). Both CARDs and PYDs belong to the death domain superfamily of signaling domains, which also includes the death-effector domains[30,31]. Thus, our findings reveal an unexpected concept for the evolution of pattern recognition in the death domain superfamily-mediated intracellular LPS-sensing pathway in vertebrate innate immunity (Supplementary Fig. 8).

In mammals, pyroptosis is considered to comprise caspase-1 and/or caspase-11 (caspase-4/5)-mediated cell death in response to certain bacterial insults[2,3,7,32]. Caspases selectively cleave the central linker in gasdermin family proteins, releasing and negating autoinhibition of the gasdermin-N domain, resulting in pyroptosis via its pore-forming activity[7,33]. Although we observed that caspy2 contributes to pyroptosis of zebrafish fibroblasts in response to both cytosolic LPS and bacterial infection (Fig. 3), it remains unknown whether this enzyme cleaves a substrate to induce pyroptosis, as in mammalian cells. Previous data have already revealed that the active N-terminal domain of the only two gasdermin family proteins identified in zebrafish, dfna5a and dfna5b, exert pore-forming effects[33]. Recently, zebrafish dfna5a (GSDME1) has been shown to possess a caspase-3 cleavage motif and induce pyroptosis following administration of chemotherapy drugs, although the role of dfna5b (GSDME2) remains unknown[33]. Thus, we speculated that dfna5a or dfna5b, or an as-yet-unidentified gasdermin family protein, fulfills the "executioner" function in fish, similar to that observed in mammals.

When caspase-4/5/11 activation by detecting cytoplasmic LPS, they trigger pyroptotic cell death and caspase-1 dependent IL-1β production[34]. In zebrafish, although the key components of the inflammasome complex assembly are conserved, the nod-like receptors are less well understood[11]. In this study, we identified the zebrafish caspy2 activation-mediated pyroptosis, both in response to cytosolic LPS and 0909I E. piscicida infection, but whether this caspy2-mediated noncanonical inflammasome activation can lead to "NLRP3 inflammasome" activation as observed in mammalian cells remain unknown. In adult zebrafish primary leukocytes, both caspy and caspy2 can cleave IL-1β in response to bacterial infection[14]. Moreover, zebrafish ASC interacts with caspy but not with caspy2, and the oligomerized form of the conserved adapter protein ASC had been shown to recruit and activate caspy, the zebrafish functional homolog of caspase-1, forming a characteristic speck structure both in vitro[10] and in vivo[35]. Thus, additional works still need to clarify whether the caspy2 mediated noncanonical inflammasome activation could trigger caspy processing, is this ASC dependent? And what's the mechanism about the zebrafish IL-1β cleavage and secretion, is this dependent on caspy2 or caspy?

Caspase-11 activation in intestinal epithelial cells plays an important role in the promotion of mucosal defense against enteric bacterial pathogens[36], indicating that pyroptosis is not only limited to monocytes, but can also occur in nonmyeloid cells. Caspase-11 exerts proinflammatory effects and mediates

gut defenses against a large variety of pathogens[3,16,36]. Mice deficient in caspase-11 exhibit enhanced Salmonella typhimurium colonization in the intestinal epithelium upon infection with this intracellular enteric pathogen, together with a reduction in histopathological signs of cecal inflammation[3,4]. Our results suggested that the noncanonical inflammasome is critical in nonmyeloid cell-mediated gut protection during 0990I E. piscicida infection in vivo. Thus, it's interesting to address the different phenotypes of mutant bacteria with that wild-type E. piscicida, which might add new findings to the molecular mechanisms in bacteria that promotes host immune recognition of replicating pathogens via noncanonical inflammasome and restricts bacterial colonization. In fish, it is generally accepted that they are exposed to substantially higher numbers of pathogens than nonaquatic vertebrates[37]. When a fish is exposed to pathogens during the mouth and gut opening stages, the intestine, a highly complex organ, plays a crucial role in the immune system[38]. However, it is not clear whether the noncanonical inflammasome exists or plays a role in fish innate immunity. In this study, we developed a bacteria-zebrafish larvae immersion infection model, and demonstrate that the zebrafish caspy2 noncanonical inflammasome governs bacterial clearance and gut inflammation in vivo (Fig. 6), which paves the way for future studies to elucidate the molecular mechanisms of the noncanonical inflammasome in lower vertebrates.

In summary, we report a unique model to study the impact of noncanonical inflammasome activation in inflammation and endotoxic shock progression with clear complementarities with the mouse model. In this model, we revealed a previously unappreciated pathway of the pyrin-like domain in cytosolic LPS sensing, which mediates noncanonical inflammasome activation in zebrafish fibroblasts. Activation of the caspy2 noncanonical inflammasome plays critical role in the clearance of bacterial infection in vivo (Fig. 7). These results provide the possibility to study inflammation and pathogen responses simultaneously in a whole organism, which should lead to the possibility of genetic and chemical screenings.

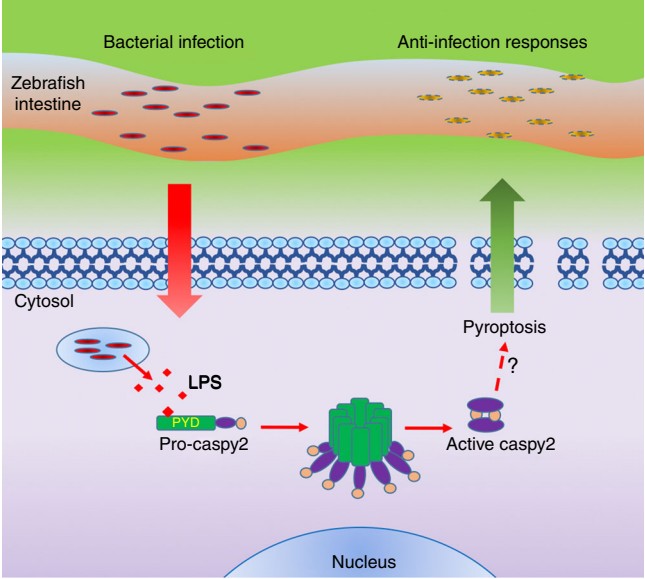

**Fig. 7** Proposed mechanism for noncanonical inflammasome activation in zebrafish. Summary of the caspy2-noncanonical inflammasome activation in zebrafish illustrating the effects of antibacterial infection effects in the zebrafish intestine. LPS lipopolysaccharide; PYD pyrin death domain

## Methods

**Zebrafish stocks and embryo collection**. Wild-type AB zebrafish were maintained in the Zebrafish Core Facility of the East China University of Science and Technology (Shanghai, China) under standard conditions[39]. Experiments conformed to regulatory standards according to protocols approved by the Animal Care Committee of East China University of Science and Technology (approval number #2006272). Zebrafish larvae were maintained until 10 dpf, at which time they were euthanized with an overdose of 4 g/L buffered tricaine (MS-222, ethyl 3-aminobenzoate methanesulfonate, Sigma-Aldrich) in accordance with ethical procedures[39] (www.zfin.org).

**Bacterial culture conditions**. Wild-type E. piscicida (EIB202, CCTCC M208068) and 0909I E. piscicida (screened from an E. piscicda gene-defined mutant library[40]) were grown in tryptic soy broth (TSB, BD Biosciences) containing 16.7 µg/mL colistin at 30 °C. For infection of cells or zebrafish larvae (as described below), bacteria were grown to the stationary phase overnight in TSB at 30 °C with aeration.

**Cell transfection and infection**. Mycoplasma free-ZF4 cells (ATCC CRL-2050™), established from 1-day-old zebrafish embryos, were grown at 28 °C in Dulbecco's modified Eagle's medium (DMEM)/F12 (Gibco) medium supplemented with 10% fetal bovine serum (FBS; Gibco) in an atmosphere containing 5% (v/v) $CO_2$. These cells were infected by replacing the culture medium with Opti-MEM (Gibco) and adding E. piscicida at a multiplicity of infection (MOI) of 50. For CTB (List Biological Laboratories) treatment, Pam3CSK4 (InvivoGen) primed cells were stimulated with 20 mg/mL CTB plus 1 mg/mL ultrapure LPS (E. coli O111:B4, InvivoGen)[22,41]. For inhibitor assays, the cells were pretreated with Z-WEHD-FMK (ApexBio), Z-YVAD-FMK (ApexBio), Ac-LEVD-CHO (Sigma), Z-VAD-FMK (Beyotime Biotechnology), Ac-DEVD-CHO (Beyotime Biotechnology), or 10 mM glycine (Sigma), respectively, for 30 min before being exposed to E. piscicida at an MOI of 50.

**Cell death and LDH release measurement**. To examine the morphological changes during cell death, cells were treated as indicated in 12-well plates and images were captured. PI (5 ng/mL) was added to the medium as an indicator of cell membrane integrity. Static bright-field images of pyroptotic cells were captured using an Olympus IX71 microscope at room temperature and processed using ImageJ software[42].

Aliquots of cellular supernatants were transferred into 96-well plates (round bottom) and centrifuged at 1000×g for 5 min. The supernatants were transferred to another 96-well plate (flat bottom), and the plate was subjected to the cytotoxicity assay using a CytoTox 96 assay kit (G1780, Promega, Madison, WI, USA) according to the manufacturer's protocol. Each sample was tested in triplicate. Cytotoxicity was normalized to Triton X-100 treatment (100% of control), and LDH release from uninfected/untreated cells was used for background subtraction.

**Caspase substrate cleavage assay**. ZF4 cells were lysed with caspase assay buffer (50 mM HEPES, 100 mM NaCl, 0.1% CHAPS, 10 mM DTT, 1 mM EDTA, and 10% glycerol, pH 7.4) and centrifuged at 21,000×g for 2 min. To analyze caspase-5-like activity, cell lysates or purified caspy2 were mixed with Ac-WEHD-AFC (BioVision) in a black 96-well plate (Costar), and the resulting fluorescence was measured using a SpectraMax M2 microplate reader (excitation at 400 nm and emission at 505 nm, Molecular Devices). Substrate cleavage was evident as an increase in fluorescence after 1 h. The activity of other caspases was assayed according to the substrate manufacturer's instructions (Beyotime Biotechnology). Briefly, chromogenic substrates of human caspase-1 (Ac-YVAD-pNA), caspase-2 (Ac-VDQQD-pNA), caspase-3/7 (Ac-DEVD-pNA), caspase-4 (Ac-LEVD-pNA), caspase-8 (Ac-IETD-pNA), and caspase-9 (Ac-LEHD-pNA) were incubated with cell lysates or purified caspy2 at a final concentration of 200 mM. Caspase activity was then determined by measuring changes in absorbance at 405 nm at 5-min intervals caused by the presence of free pNA hydrolyzed from the substrates.

**Generation of KO cells using CRISPR/Cas9**. A CRISPR/Cas9 gRNA expression vector (pSpCas9(BB)-2A-GFP, #48138) was obtained from Addgene. The caspy2-KO target sequences were 5′-GGCGTCGAACCCATTCCTCG-3′ and 5′-GAATA AGGACCGTCAGGAT-3′. To generate caspy2-KO ZF4 cell lines, DNA-In CRISPR Transfection Reagent (MTI-GlobalStem) was used to transfect the plasmids. Two to three days later, GFP-positive cells were sorted for culture by flow cytometry, and caspy2-KO clones were identified by immunoblotting with rabbit-anti-caspy2 antibodies (1:1000; custom-made; Genscript).

**HeLa cell reconstitution experiments**. Mycoplasma free-wild-type, $CASP4^{-/-}$, and $GSDMD^{-/-}$ HeLa cells were grown at 37 °C in DMEM supplemented with 10% FBS in an atmosphere containing 5% (v/v) $CO_2$. $CASP4^{-/-}$ HeLa cells were transduced with pCDH expression plasmids for caspy2, caspy2 (C296A), or caspy2 (ΔPYD). After 3 days, puromycin-resistant transduced cells were collected. Indicated cells were primed with Pam3CSK4 for 4 h, and stimulated with CTB plus LPS for 12 h. Levels of caspy2 proteins were determined by immunoblotting. $CASP4^{-/-}$

HeLa cells were transduced with pCDH expression plasmids for HA-tagged caspase-11, caspase-11 (C254S), or caspase-11 (ΔCARD). After 3 days, puromycin-resistant transduced cells were collected. Indicated cells were primed with Pam3CSK4 for 4 h, and stimulated with CTB plus LPS for 12 h. Levels of caspase-11 proteins were determined by immunoblotting using rabbit-anti-HA antibodies (1:5000; GeneTex).

**Purification of proteins**. Mycoplasma free-HEK293T cells were plated in a 100-mm Petri dish 24 h before transfection. Cells were then cotransfected with pCDH expressed HA-tagged caspy2, caspy, caspase-1, or caspase-11. Cells were collected and lysed in a buffer containing 20 mM Tris, 100 mM KCl, 0.1% NP-40, 1 mM EDTA, 1 ml EDTA, 10% glycerol, 10 mM tetrasodium pyrophosphate and protease inhibitor cocktail (Roche Molecular Biochemicals) for 30 min. The lysates were divided and incubated with Pierce Anti-HA magnetic beads (Thermo Fisher Scientific), 6 h later, the HA beads were washed three times with cell lysis and wash buffer (20 mM Tris, 150 mM KCl, 0.5% NP-40, 1 mM EDTA, 1 ml EDTA, 10% glycerol, 10 mM tetrasodium pyrophosphate and protease inhibitor cocktail). The HA-tagged proteins were then eluted using an HA peptide (Apexbio) according to the manufacturer's protocol.

**LPS pulldown assay**. Biotinylated-ultrapure LPS and Biotinylated-Pam3CSK4 from E. coli O111:B4 (InvivoGen) were incubated with Pierce™ Streptavidin Magnetic Beads (Thermo Scientific) for 6 h. The beads were then centrifuged at 800×g for 1 min and the supernatant was removed. One milliliter of purified HA-tagged caspy2, caspy2 (C296A), caspy2 (ΔPYD), caspy2 PYD, caspy, or caspy PYD protein in HEK293T cells was mixed with LPS or Pam3CSK4-bound beads at 4 °C overnight, after which the beads were again centrifuged at 800×g for 1 min and the supernatant was removed. After three washes with PBST (phosphate buffered saline containing 0.05% Tween 20), the beads were boiled with sample loading buffer. The precipitates were subjected to SDS-PAGE and subsequently transferred to polyvinylidene difluoride (PVDF) membranes by electroblotting.

**ELISA-based LPS binding assay**. The LPS-binding activities of purified HA-tagged caspy2, caspy2 (C296A), caspy2 (ΔPYD), caspy2 PYD, caspy, caspy PYD, caspase-1, and caspase-11 proteins in HEK293T cells were measured by ELISA. Briefly, a 96-well microtiter plate (Costar) was coated with 20 mg E. coli O111:B4 LPS (Sigma-Aldrich) in 100 µl of carbonate-bicarbonate buffer (pH 9.6) per well, and incubated at 4 °C overnight. The wells were then washed four times with PBST (containing 0.05% Tween 20) and blocked with 200 µl of PBS containing 3% bovine serum albumin at 37 °C for 1 h. The plate was thoroughly washed, and 100-µl twofold serial dilutions of purified protein were added to each well and incubated at room temperature for 2 h. The plate was again thoroughly washed, and a mouse anti-HA tag monoclonal antibody (diluted 1:2000, Abcam) was added and incubated at room temperature for 1 h to detect the bound purified proteins.

**Caspy2 foci staining and caspy2 oligomer cross-linking**. ZF4 cells were seeded on coverslips at a density of $2 \times 10^5$ cells per well in 24-well plates and cultured overnight. After infection with indicated bacterial strains, cells were washed with PBS and fixed with 4% paraformaldehyde for 15 min, permeabilized with 0.1% Triton X-100 for 5 min, and blocked with 10% FBS in PBS for 20 min. Cells were then incubated with anti-caspy2 antibodies (1:1000; custom-made; Genscript), followed by goat antirabbit IgG (H + L) secondary antibody, oregon green 488 (O-11038, Thermo Fisher Scientific) and 2-(4-amidinophenyl)-1H-indole-6-carboxamidine (C1005, Beyotime Biotechnology). Images were acquired using a Leica DMI3000B inverted fluorescence microscope.

For caspy2 oligomer cross-linking, ZF4 cells were plated on 12-well plates and stimulated as indicated. Cells were lysed with PBS buffer containing 0.5% Triton X-100, and the lysates were centrifuged at 6797×g for 15 min at 4 °C. Supernatants were transferred to new tubes (Soluble fractions). The Triton X-100-insoluble pellets were washed with PBS twice and then suspended in 200 µl PBS. The pellets were then cross-linked at room temperature for 30 min by adding 4 mM dextran sulfate sodium. The cross-linked pellets were centrifuged at 6797×g for 15 min and dissolved directly in SDS sample buffer.

**Blue native PAGE**. Blue native gel electrophoresis was performed using the Bis–Tris native PAGE system as previously described[6,43]. ZF4 cells were plated on 12-well plates and stimulated as indicated. Cells were washed once with cold PBS and then lysed in ice-cold native lysis buffer (20 mM Bis–Tris, 500 mM ε-aminocaproic acid, 20 mM NaCl, 10% (w/v) glycerol, 0.5% digitonin, 0.5 mM $Na_3VO_4$, 1 mM PMSF, 0.5 mM NaF, 1× EDTA-free Roche protease inhibitor cocktail, pH 7.0) for 15 min on ice. Lysates were clarified by centrifugation at 20,000×g for 30 min at 4 °C and analyzed without further purification steps. Cell lysates were equalized after quantification of total protein using the BCA protein assay (Pierce), and then separated by 4–12% blue native PAGE. Native gels were incubated in 10% SDS solution for 5 min before transfer to PVDF membranes (Millipore), followed by conventional western blotting.

**Morpholino design and analysis**. An morpholino oligonucleotide (Gene Tools) was designed to target a site in *caspy2* to block its translation (5′-CTGGGTAAT ATCCTCCATTTTCTGT-3′), and a corresponding 5-base mismatch oligonucleotide was used as a control (5′-CTGGcTAATATgCTgCATTTTgTcT-3′). Embryos were injected as described previously[44]. MOs were used at 0.5 ng per embryo, and the results of experiments using three batches of embryos, performed and analyzed independently, are shown. Knockdown of caspy2 expression was verified by immunoblotting with rabbit-anti-caspy2 antibodies (1:1000; custom-made; Genscript).

**Immunoblotting**. Lysates were subjected to SDS-PAGE before being transferred to PVDF membranes (Millipore) by electroblotting. For immunoblotting, the rabbit-anti-caspy2 antibody (1:1000; custom-made; Genscript), or rabbit-anti-HA antibody (1:5000; GeneTex) were used. The same amounts of lysates were leaded on each gel and probed with anti-β-actin antibodies (1:5000; Abcam) as a loading control.

**Whole-mount in situ hybridization**. For whole-mount in situ hybridization, digoxigenin-labeled caspy2 antisense RNA probes were synthesized from full-length cDNA sequences using an in vitro transcription kit (Promega). In situ hybridization and development of whole-mount zebrafish embryos were performed as described previously[10].

**Bacterial immersion infection model**. For immersion infection of zebrafish larvae, bacteria were prepared as described in the Bacterial Culture Conditions subsection. Larvae at 5 dpf were randomly divided and immersed in PBS or PBS containing $10^5$ CFU/ml 0909I *E. piscicida* for 2 h, before being allowed to recover for 15 min in a petri dish containing fresh E3 medium. Subsequently, they were transferred to 10-cm dishes, with approximately 50 larvae in 15 ml of E3 medium per dish, and incubated at 28 °C. The mortality rate and bacterial colonization were then observed at different time points. For histopathology, embryos were injected with caspy2- or control-MO at the one-cell stage, and fixed in 4% paraformaldehyde at 72 hpi. Whole larvae were dissected and tissues fixed in formalin and embedded in paraffin using standard methods[39]. Tissue sections were evaluated microscopically for signs of inflammation following hematoxylin and eosin staining[39].

**Lethal sepsis model**. Zebrafish larvae at 5 dpf were randomly divided and immersed in 100 μg/ml *E. coli* O111:B4 LPS (Sigma) for 2 h, before being allowed to recover for 15 min in a petri dish containing fresh E3 medium. Subsequently, they were transferred to 10-cm dishes, with approximately 50 larvae in 15 ml of E3 medium per dish, and incubated at 28 °C. Survival rates were calculated at the indicated time points. The RNA of infected zebrafish larvae was extracted using an RNA isolation kit (Tiangen, Beijing, China). One microgram of each RNA sample was used for cDNA synthesis with the FastKing One Step RT-PCR Kit (Tiangen) and quantitative real-time PCR (qPCR) was performed on an FTC-200 detector (Funglyn Biotech, Shanghai, China) by using the SuperReal PreMix Plus (SYBR Green) (Tiangen). The expression level of the genes ecoding IL-1β, TNF-α, IL-6, IL-8, IL-10, and IFN-γ (Supplementary Table 1) were assessed for three biological replicates, and the data for each sample were expressed relative to the expression level of the gene encoding beta-actin gene by using the $2^{-\Delta\Delta CT}$ method[39].

**Statistical analysis**. Statistical analysis was performed using GraphPad Prism (GraphPad Software). All data are representative of at least three independent experiments and are presented as means ± SD. Two-group comparisons were made using Student's *t* test, and one-way analysis of variance was used to assess differences among multiple groups. Differences in larvae survival were assessed using the log-rank (Mantel–Cox) test. *p* Values less than 0.05 were considered to indicate statistical significance (*$p < 0.05$, **$p < 0.01$).

**Data availability**. All relevant data are available from the authors on request and/or are included with the manuscript (as figure source data or supplementary information files).

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

## Acknowledgments

We thank Dr. Feng Shao from NIBs for sharing *CASP4*$^{-/-}$ and *GSDMD*$^{-/-}$ HeLa cells, and providing useful advice. This work was supported by the National Natural Science Foundation of China (Nos. 31472308 (Q.L.) and 31430090 (Y.Z.)) and the Fundamental Research Funds for the Central Universities (No. 222201714022 (D.Y.)). Dahai Yang was supported by the Young Elite Scientists Sponsorship Program by CAST No. 2016QNRC001, Shanghai Pujiang Program No. 16PJD020, Shanghai Chenguang Program No. 16CG33, and Talent Program of School of Biotechnology in East China University of Science and Technology. Ruilin Zhang was supported by the National Natural Science Foundation of China (No. 31571492).

## Author contributions

D.Y. and Q.L. were responsible for project conception, data analysis, and manuscript editing; D.Y. performed the majority of experiments, assisted by S.C., X.Z., Z.W., W.X., J. T., T.H., M.H., W.W., and G.Z.; X.Z. helped with the in vivo experiments. Q.W., R.Z., and Y.Z. provided expert advice and critically reviewed the manuscript. The manuscript was written by D.Y. and Q.L., who also supervised the study.

## Additional information

**Competing interests:** The authors declare no competing interests.

