## [Peer Review File · Nature Communications]

Reviewers' comments:

Reviewer #1 (Remarks to the Author):

Yang et al. study zebrafish Caspy2, which encodes a caspase of unknown function. The authors use infection with *Edwardsiella piscicida* that has been engineered in an unspecified manner to change its virulence somehow. This strain is used in vitro and in vivo, showing that the infection triggers cell death via Caspy2 that is morphologically similar to pyroptosis seen in mammalian cells. Caspy2 has substrate specificity similar to human caspase-5, which is an LPS sensor. Caspy2 was shown to immunoprecipitate and bind in an ELISA with LPS. Finally, zebrafish immersed in these bacteria were more susceptible to infection if Caspy2 was knocked down with a morpholino, but were paradoxically more resistant to the bacteria at high dose infection.

The authors may be right that Caspy2 detects LPS, but their data is not fully supported. Critical experimental details are omitted, including the nature of the engineered bacteria used in the study, preventing a full evaluation of the data presented. The LPS binding assays require numerous additional controls. The in vivo data are paradoxical. In summary, this is an interesting manuscript that proposes the hypothesis that Caspy2 is a cytosolic LPS sensor. The likelihood of this being true is high, however the data are in a very preliminary stage not suited for publication yet.

Significant points.

The authors did not explain the origin of 09091 *E. piscicida*. Is this strain able to lyse the vacuole and enter the cytosol? If not, how does the LPS enter the cytosol? I do not see an explanation of what the engineering of this strain entailed in the supplementary figure, and the methods section also does not explain, nor reference another paper. Google and pub med searches for 09091 *Edwardsiella* received no matches. Thus, I am unable to evaluate and critique the in vitro and in vivo infection data using this strain with regards to its physiological relevance. Once the authors explain what this strain is, and why the WT bacteria do not trigger Caspy2, then I may have additional comments related to the use of this strain and how the authors interpret their data.

Authors should include a phylogenetic tree of human Casp1-4-5 and mouse Casp1-11 with the zebrafish Caspy1-2 to show if there is any phylogenetic relation. Human Casp4 and Casp5 cluster with mouse Casp11 and the casp1 cluster together in such trees, so one would expect the zebrafish caspases to correctly fit into the tree.

Figure 2a and 2b are not annotated correctly. it is not "Control, WT, and KO1, and KO2". It should be Control, Control, KO1, KO2. And then also Uninfected, Infected, Infected, Infected.

Why is there so much more Caspy2 in the infected control lane? One assumes that the infection is 2h MOI 50 as in Fig. 1 which is not long enough for transcriptional induction of Caspy2 I would think.

The methods do not clearly state whether 2h MOI 50 is used throughout. It is stated in in Fig. 1 but not the methods or other figures.

LPS pulldown results in Fig. 3a are quite faint bands. additional controls are required for a pulldown with LPS – this is not expected to be a clean simple pulldown like other protein-protein interactions. The LPS exists in large micelles that are, of course, lipid. Thus, they are likely to be quite sticky in a non-specific manner, necessitating the inclusion and addition of many controls. The authors should include Caspy1 as a negative control in addition to the delta PYD Caspy2. They should also include bead controls without LPS. They should also include some other lipid-type control that should not interact; in this regard, invivoGen sells biotinylated Pam3CSK4 (<http://www.invivogen.com/pam3csk4-biotin>). Tetra-acylated LPS would also be a welcome

control. These same controls should be integrated into the LPS ELISA, as that risks the same lipid-stickiness caveats. These controls are even more important since Caspy2 can polymerize, which would tend to make it pull down as a particulate if it activated nonspecifically – thus the delta PYD mutant is an insufficient control as the full length protein could precipitate whereas the delta PYD mutant would be soluble. Finally, PYD alone should also bind LPS, whereas the equivalent domain from Caspy1 should not.

Fig. 3c. One cannot compare an intact cell to a pyroptotic cell and make these conclusions. It appears that most of the green signal has been lost from the cell, consistent with pyroptosis lysing the cell and releasing the protein. The authors need to repeat this under conditions where there is no pyroptosis. I suggest doing the experiment in 10mM glycine to inhibit pyroptosis, as well as trying it in the presence of a caspase-inhibitor. DIC/phase images should be shown alongside to show lack of pyroptosis.

The methods mention purified Caspy2, but I did not see methods on how this was purified, simply an HA tag, or immunoprecipitated as suggested by Supp Fig. 4? Figure 3 legend states that Caspy2 proteins were “prepared” – what does that mean? Methods section says that the proteins were purified but does not say how this was done. Methods must be included.

Fig. 5a and 5b. mRNA says there is a 30 fold induction of Caspy2 expression in Fig. 5a. But I see almost no difference in total Caspy2 protein in Fig. 5b. How do the authors reconcile this discrepancy?

The processing in Fig. 5b is quite weak. If one is willing to accept weak band, then the WT bacterial infection also causes the appearance of new weak band at around 35 kD. Thus, the data is questionable for its relevance.

You cannot say that the “control MO larvae were unaffected (Fig. 5c)” when one of them died and all had bacterial burdens in Fig. 5d.

Figure 5d is plotted inappropriately. There are dead fish in this experiment, the numbers of which are different starting at the earliest time point. Were the CFU burdens from dead fish included in the graph? If so, these carry the caveat that the fish was dead and bacteria will be replicating with no check from the fish immune system. If they were excluded, then statistical analysis cannot be performed because the highest points in the graph were excluded (assuming fish with highest burdens were the ones that died, which is likely). In either case, the dead fish need to be graphically represented on the graph. One could mark these with an X through the symbol, or instead of the symbol. If dead fish are excluded, X could be added above the axis of the graph to show that a data point is missing. In either case, statistical analysis will be difficult to perform.

Fig. 5c vs. Fig. 5i shows that the Caspy2 MO knocked down fish have greater mortality when infected with a low dose of 10^5 than they do at a high dose of 10^9 . This is biologically paradoxical, and leads one to wonder about the physiologic relevance of the infectious model.

Minor aspects.

The introduction should mention how many caspase genes are present in the zebrafish genome.

“...and the formation of the p20 cleaved form of Caspy2 was detected in wild-type LPS-transfected ZF4 cells, but was impaired in caspy2-KO cells (Fig. 3e).” This is incorrectly stated. Of course the Caspy2 p20 is absent from knockout cells because it was knocked out, not that presence of the p20 was “impaired”. Also, the p20 is so weak as to lead one to wonder whether cleavage has much meaning for Caspy2. Cleavage of caspase-11 has been a poor marker for its catalytic activity.

Reviewer #2 (Remarks to the Author):

Review of Yang et al. for Nature Communications

Summary:

In this manuscript, Yang and colleagues examined the role of the zebrafish caspase family member Caspy2. Based on the homology of Caspy2 to mammalian Caspase 11/5 and its ability to cleave canonical Caspase 11 substrate, authors conclude that Caspy2 might represent a conserved zebrafish equivalent of non-canonical inflammasome in zebrafish. They provide data that suggest that Caspy2 has the ability to interact with LPS and suggest that this involves its PYD-domain. Furthermore, the data show that Caspy2 can somehow initiate pyroptotic-like cell death both in human HeLa cells and in zebrafish fibroblast ZF4 cell line, yet the mechanism remains unclear. Finally, the authors also perform in vivo study of Caspy2 expression and activation, involving Caspy-2-MO larvae, indicating a role for Caspy-2 in protecting fish larvae from bacterial infection in the gut. Moreover, they also establish and provide important data on Caspy2 function in a novel model of LPS-induced mortality in larvae.

Although homology of Caspy2 to mammalian Caspase 5/11 and its ability to cleave a Caspase 11/4/5 substrate has been demonstrated in several studies before, it was so far not shown whether Caspy2 is indeed a functional homolog of Caspase4/5/11 functions in zebrafish model. The study also provides intriguing observation that PYD domain of Caspy2 might be important for its LPS binding and activation, which, considering previously demonstrated LPS-binding activity of mammalian CARD-containing Caspase 11, suggests that this function might be conserved for the different death fold superfamily domains, and that PYD and CARD domain functions might have been swapped during evolution of vertebrate immune system. However, more experiments are necessary to fully support this claim and to further characterize caspy-2 expression/induction and the Zebrafish non-canonical pathway. From an experimental aspect, the quality and presentation of the data needs improvement, and essential controls are still lacking (see below). In addition, the Figure legends and materials and methods section are poorly written and in many occasions, but not least, I am worried whether sound scientific practises were applied when generating the data based on discrepancies in the full western blots presented in Suppl. Figure 10 (for details see below).

General remarks:

1) The authors do not at all explain their choice of the bacterial strains used in the study, and the genetic difference between EIB202 and O9091 strains and do not provide evidence that this strain specifically activates only "non-canonical inflammasome" but not canonical pathway in zebrafish cells. How are these bacteria entering the cytosol of ZF4 cells upon infection, and how is this related to the genetic difference between EIB202 and O9091 strains.

They do not provide a data indicating whether the canonical inflammasome pathway components are expressed or not in ZF4 cells, therefore it is difficult to assess potential canonical inflammasome contribution to phenotypes observed both in vitro and in vivo. It is also known from the literature that non-canonical inflammasome activation can lead to NLRP3 inflammasome activation in mammalian cells. Is this also a case in Zebrafish cells?

2) The authors claim that Caspy2 is strongly upregulated upon bacterial infection (see Fig. 2), and, later, by LPS treatment (Fig. 3e). However, this data is conflicting with the in vivo data presented later Figures, where Caspy2 mRNA seems to be present constitutively in gills and gut (Fig. 4), and where Caspy-2 is expressed even in uninfected conditions (0h p.i., Figure 5b) and expression

doesn't increase with time (Fig. 5b). The authors do not provide any explanation to this. Furthermore, in Figure 3e the authors present blots of cells that were apparently primed with LPS for 4 and then differentially treated. Here Caspy-2 expression is only observed when either LPS is used or LPS-CTB. This is confusing, since either long priming is necessary (4h +), or LPS needs to reach the cytosol to prime Caspy-2 expression.

Therefore, it will be necessary to clarify these discrepancies and define under which conditions Caspy-2 is induced and what pathway drives its expression.

3) The Western blot image in the Figure 2b shows the processing of Caspy2, with several minor bands appearing after infection. However, the authors do not provide any data indicating whether this processing is necessary for Caspy2 activation and pyroptosis, although they indicate it on the scheme (Supplementary figure 6). At the moment it is disputed if processing is necessary for activation of mouse Caspase-11.

Thus, the authors need to identify autoproteolysis sites, and confirm that autoprocessing is an essential prerequisite for caspy2 activity (pyroptosis, WEHD activity) by mutating potential cleavage sites.

4) What are the components of the noncanonical inflammasome in Zebrafish? In mice Caspase-11 only induces cell death via GSDMD, but can't process IL-1b. IL-1b processing is however observed after Casp-11 activation due to the engagement of the NLRP3-ASC-Casp-1 inflammasome that responds to GSDMD induced potassium release. Does Caspy-2 process Zebrafish IL-1b, or only Caspy-1. IS IL-1b production observed during infection with *E. piscida*, and if yes, is it Caspy-2 or Caspy-1 dependent. Following these leads it should also be possible to identify the NLRP3 equivalent in Zebrafish.

5) The data on Caspy-2 PYD binding LPS are too preliminary and require additional experiments. The authors need to show that the Caspy-2-PYD itself has the ability to interact with LPS, and ideally that transferring the Caspy-2 PYD on Caspy-1 transfers the ability to bind LPS (similar experiments were done by Feng Shao in Shi et al. Nature 2014). Furthermore, it needs to be shown that Caspy-1 does not bind to LPS nor other PYD-containing proteins (e.g. ASC). Additional ligands needs to be tested as well, as done by Shi et al., such as biotin-Pam3CSK4 or MDP.

6) The authors propose that oligomerization of Caspy-2 is induced by LPS binding and is required for Caspy-2 activation and auto-processing. Yet, no evidence is shown, except in Figure 3c, where somekind of Caspy-2 speckles can be observed.

Oligomerization of Caspy-2 upon LPS binding should be assessed ideally both with purified protein, by performing size-exclusion analysis of Caspy-2 with and without addition of LPS, and of the endogenous Caspy-2, by detecting the oligomeric form.

The punctate structures in Fig. 3c alone do not provide an evidence of direct interaction between single Caspy2 molecules. It also cannot be ruled out that the punctate cytoplasmic structures that authors observe arise as a treatment artefact or, alternatively, contain additional partners involved in LPS binding and pyroptosis initiation. Authors also claim that there was a significant difference in number of these structures observed upon infection with two strains, however, this data is not included to the manuscript (quantification).

7) Authors do not provide enough data on Caspy2 role in their Zebrafish larvae septic shock model. They do not state what physiological and morphological changes occur in larvae, and whether Caspy2 activation leads to the cytokine processing and release. Therefore, the relevance of this model to existing septic shock models in higher vertebrates remains unclear.

8) The western blots shown in Figure S10 are a major point of criticism. A) In general it has the appearance that the authors just cut out what they liked from these blots, no information is for example provided on what the other bands are from these blots. B) More seriously, I am quite shocked by the fact that the b-actin controls do not appear to arise from the same blot as the Caspy-2 blots for example. Fig. 3e provides one example where Caspy-2 data were obtained from 2 blots, while a 3rd blots with some random bands apparently serves as the b-actin control. To convince me that equal amounts were loaded, the authors should strip the Caspy-2 blots and re-blot for b-actin. C) Most serious: The blot shown for Fig. S9a shows the timecourse of Caspy-2 induction in normal larvae (lane 1-6) and Caspy-2-MO larvae (lanes 7-12). The crossreactive band at high molecular weight shows can serve to identify the different lanes (i.e. 0.5, 1, 3 and 5 dpf). Shockingly the blot suggests that the authors selected lanes 2-5 to show induction of Caspy-2 in

Wt larvae, while showing lanes 7-10 in the Caspy-2 MO larvae. Based on the upper band, I think that lanes 8-11 should have been selected, which would show that there is no difference in Caspy-2 induction between the WT and presumed Caspy2-MO's, which would question the validity of all in vivo data in the study! Furthermore, the blot suggests that the authors intentionally do not show a later timepoint at which Caspy-2 is expressed in Caspy-2-MOs! Since the lanes are not labelled and no more explanation is given for the blots shown in S10, I would ask the authors to comment on my remarks.

Also, b-actin blots in S9a cannot possibly be from the same membrane as the Caspy2 blots.

Additional comments:

In the introduction, authors should elaborate more on general importance of inflammasome pathway in vertebrates, as well as advantages of Zebrafish as a model to study innate immune response. The objectives of the study need to be stated more clearly.

Fig 1: The authors observe high levels of cytotoxicity (up to 20%) in their controls in Figures 1,2,3. However, they do not provide an explanation to this phenomenon. For example it is unclear how the LDH levels were calculated. Normally, % LDH release is normalized to uninfected or mock-infected samples. This does not appear to be the case, as in Fig. 1a, mock-infected samples show already 20% of LDH release. What was the zero-value that was used to calculate the percentage of LDH release?

Fig. 2b: Confirm that CRISPR-KO of Caspy-2 does not change the activation of canonical inflammasomes, which could account partially for the cell death.

Fig. 3a. Equal amounts of protein need to be loaded in the input, since like this it is impossible to judge if really no Caspy2-Dpyd has been pulled down.

Fig. 3a: There is no description how the protein was purified. If purified from bacterial sourced, we can expect it to bind LPS already, which would interfere with the LPS PD>

Fig. 3a: What is the second band in the 4th lane of the second blot that runs at 24 kDa?

Fig. 3b: How was the ELISA used in this experiment validated. Data need to be shown that it works with other LPS-binding proteins and a negative control (e.g. Casp-1) needs to be added as well.

Fig. 3c: I am intrigued by the Caspy-2 speckles. No comparable structures can be seen for mouse Caspase-11 during LPS TF or bacterial infections. Are these oligomers of Caspy-2, or Caspy-2 binding to the bacterial surface? Can these oligomers be detected by SDS-PAGE/WB after X-linking?

Fig. 3e: This experiments needs to be repeated under conditions where cells have been primed to express Caspy-2 (Pam3CSK4 priming?). Based on what is shown here, we cannot exclude that CTB alone would activate Caspy-2, since no Caspy-2 is present in lane 3.

Fig. 3f: Additional controls necessary such as complementation of Casp4-/- HeLa with Casp-4 and also Casp-11.

Fig. 3g: compared to WT Caspy2, D296A and Dpyd are poorly expressed. Needs to be repeated with equal amounts.

Fig. 3f/g: What is the contribution of human Casp5 in these assays?

Fig. 4b: What is the 42 kDa band that appears in lane 2-4. Is it a splice variant of Caspy-2? And why does it disappear with time?

Another issues: markers seemed to be different in the full blots (Fig. S10).

Fig. 4c: there is a strong In-situ hybridisation signal for the embryo at 12 hpf, which can be viewed as a high level of Caspy2 mRNA. This is controversial to the data from the panels a and b of the same figure.

Fig 4: It is known from literature that mRNA level does not always predict protein expression level. Therefore, it could be interesting to check Caspy2 protein expression in embryos during different developmental stages, as well as at different time points before and after infection, using whole embryo immunofluorescence approach.

Fig5a: What pathway is engaged that induces Caspy-2 in vivo?

Fig.5b: Why does the infection with EIB202 also induce the processing of Caspy-2?

Why is there no induction at protein level with time? And Why is Caspy-2 already made in lane (0 hpi)?

Fig. 5i: On what cells does LPS act and how does it enter the larvae? How does LPS reach the cytosol of cells in vivo to cause Caspy-2 activation?

Suppl. Fig. 4: I am confused by this figure. The authors claim that LPS is the ligand that activates Caspy-2, yet they observe its activation in cell extract of transfected cells, that did were neither infected nor LPS transfected.

Suppl. Fig. 9a: Here again a band is seen on the blots (at ~40 kDa) that is not commented on at all. Intriguingly it is suppressed by Caspy-2 MO, thus unlikely to be a cross-reactive band.

Supplementary fig, 9a, wt control is missing.

Page 8, lines 150-153: This needs reformulating, since the data suggest that Caspy-2 has a function similar to Casp4 rather than Casp5. Please provide the references that show that Casp4 or Casp5 CARD binds LPS. I am not aware of such papers.

We would like to thank the editor and the reviewers for the comments and insightful suggestions on our manuscript, which have greatly helped us to improve our study. Based on these comments, we have performed new experiments and clarified certain statements in our revised manuscript. Our point-to-point response to the reviewers is provided below:

Reviewer #1 (Remarks to the Author):

Yang et al. study zebrafish Caspy2, which encodes a caspase of unknown function. The authors use infection with *Edwardsiella piscicida* that has been engineered in an unspecified manner to change its virulence somehow. This strain is used in vitro and in vivo, showing that the infection triggers cell death via Caspy2 that is morphologically similar to pyroptosis seen in mammalian cells. Caspy2 has substrate specificity similar to human caspase-5, which is an LPS sensor. Caspy2 was shown to immunoprecipitate and bind in an ELISA with LPS. Finally, zebrafish immersed in these bacteria were more susceptible to infection if Caspy2 was knocked down with a morpholino, but were paradoxically more resistant to the bacteria at high dose infection.

The authors may be right that Caspy2 detects LPS, but their data is not fully supported. Critical experimental details are omitted, including the nature of the engineered bacteria used in the study, preventing a full evaluation of the data presented. The LPS binding assays require numerous additional controls. The in vivo data are paradoxical. In summary, this is an interesting manuscript that proposes the hypothesis that Caspy2 is a cytosolic LPS sensor. The likelihood of this being true is high, however the data are in a very preliminary stage not suited for publication yet.

Significant points.

The authors did not explain the origin of 0909I *E. piscicida*. Is this strain able to lyse the vacuole and enter the cytosol? If not, how does the LPS enter the cytosol? I do not see an explanation of what the engineering of this strain entailed in the supplementary figure, and the methods section also does not explain, nor reference another paper. Google and pub med searches for 0909I *Edwardsiella* received no matches. Thus, I am unable to evaluate and critique the in vitro and in vivo infection data using this strain with regards to its physiological relevance. Once the authors explain what this strain is, and why the WT bacteria do not trigger Caspy2, then I may have additional comments related to the use of this strain and how the authors interpret their data.

Reply: The reviewer raised a valid question. 0909I *E. piscicida* was screened from the transposon insertion mutant library, which was generated by conjugation between EIB202 *E. piscicida* (recipient) and SM10 λ pir/pMar2xT7 (transposon donor) (Yang et al., mbio, 2017, 8: 5e01581-17). From our

unpublished data, we have demonstrated that 0909I is the hemolysin (ethA) overexpression strain. Compared with EIB202 *E. piscicida* infection, 0909I infection would significantly promote caspase-4 dependent pyroptosis and IL-18 secretion in intestinal epithelial cells (IECs), as well as in HeLa cells (Supplementary Fig. 1). According to these evidences, we found that the overexpressed-hemolysin internalizes into cells via binding to OMVs and promotes rupture of OMV-containing vesicles, thereby releasing OMV-derived LPS into the cytoplasm and eventually triggering significant activation of non-canonical inflammasome. The relevant data have been submitted as a separate manuscript in parallel. If required, we can send the manuscript to the reviewer for reference.

In this work, based on the results in mammalian cells, we took advantage of 0909I *E. piscicida* as a tool to infect zebrafish fibroblasts in our manuscript, trying to find whether the non-canonical inflammasome activation exists in lower vertebrates and further to find which protein is the LPS-sensing caspase. And our results showed that 0909I *E. piscicida* infection, but not EIB202 *E. piscicida*, triggers significant Caspy2 activation in Zebrafish fibroblasts (revised Fig. 2), which was responsible for pyroptosis in zebrafish non-myeloid cells (revised Fig. 2).

Authors should include a phylogenetic tree of human Casp1-4-5 and mouse Casp1-11 with the zebrafish Caspy1-2 to show if there is any phylogenetic relation. Human Casp4 and Casp5 cluster with mouse Casp11 and the casp1 cluster together in such trees, so one would expect the zebrafish caspases to correctly fit into the tree.

Reply: We agree with the reviewer's suggestions. Accordingly, we constructed a phylogenetic tree of human caspase-1/4/5, mouse caspase-1/11 with zebrafish caspy and caspy2, but found that the zebrafish caspases are distantly related to the caspases (Figure R1). We assume that it is because the caspases in zebrafish contains pyrin like domain at the N-terminal site (Supplementary Fig. 3; Masumoto J., et al., J. Biol. Chem. 2003, 278: 4268–4276). We hypothesized that this difference might have played a role during evolution, which needs to be the subject of future analysis.

Figure R1. Phylogenetic tree of the caspases family of proteins in human, mouse, and zebrafish. ClustalW alignment was carried out to generate the phylogenetic tree by using the 'Neighbor Joining' method.

Figure 2a and 2b are not annotated correctly. it is not "Control, WT, and KO1, and KO2". It should be Control, Control, KO1, KO2. And then also Uninfected, Infected, Infected, Infected.

Reply: Thanks for the reviewer's constructive suggestion. We have corrected them in revised manuscript.

Why is there so much more Caspy2 in the infected control lane? One assumes that the infection is 2h MOI 50 as in Fig. 1 which is not long enough for transcriptional induction of Caspy2 I would think.

Reply: According to the reviewer's comments, we have repeated the experiments with a more specific rabbit anti-caspy2 antibody, and included the EIB202 *E. piscicida* infection samples as a control. Indeed, caspy2 was not constantly expressed in zebrafish fibroblasts. When we infected with *E. piscicida*, caspy2 was significantly induced, and the activation was detected in 0909I *E. piscicida*-infected cells.

The methods do not clearly state whether 2h MOI 50 is used throughout. It is stated in in Fig. 1 but not the methods or other figures.

Reply: The reviewer raises a valid question. In our manuscript, we did use *E. piscicida* to infect cells for 2 hpi at the MOI of 50 throughout the manuscript. We have corrected them in revised Figure legends and in the Methods.

LPS pulldown results in Fig. 3a are quite faint bands. additional controls are required for a pulldown with LPS – this is not expected to be a clean simple pulldown like other protein-protein interactions. The LPS exists in large micelles that are, of course, lipid. Thus, they are likely to be quite sticky in a non-specific manner, necessitating the inclusion and addition of many controls. The authors should include Caspy1 as a negative control in addition to the delta PYD Caspy2. They should also include bead controls without LPS. They should also include some other lipid-type control that should not interact; in this regard, invivoGen sells biotinylated Pam3CSK4 (<http://www.invivogen.com/pam3csk4-biotin>). Tetra-acylated LPS would also be a welcome control. These same controls should be integrated into the LPS ELISA, as that risks the same lipid-stickiness caveats. These controls are even more important since Caspy2 can polymerize, which would tend to make it pull down as a particulate if it activated nonspecifically – thus the delta PYD mutant is an insufficient control as the full length protein could precipitate whereas the delta PYD mutant would be soluble. Finally, PYD alone should also bind LPS, whereas the equivalent domain from Caspy1 should not.

Reply: Thank you for the constructive suggestions. Accordingly, we repeated the pulldown assays to test the LPS-binding affinity and specificity of caspy2 and caspy in zebrafish (revised Fig. 3). Biotinylated-ultrapure LPS efficiently precipitated caspy2, caspy2 (C296A), and caspy2 PYD, but not caspy2 (Δ PYD), caspy, or caspy PYD (Fig. 3a). In addition, we used the biotinylated-Pam3CSK4 as the control, and caspy and caspy2 could not be detected from *in vitro* pulldown assays.

In addition, in enzyme linked immunosorbent assays (ELISAs) of LPS-binding activity, we added the controls according to reviewers' comments. Consistent with caspase-11 did, caspy2, caspy2 (C296A) and caspy2 PYD showed strong affinity to immobilized LPS, with the number of LPS-caspy2 complexes increasing in a concentration-dependent manner (revised Supplementary Fig. 6). However, caspy2 (Δ PYD), caspy, caspy PYD, and caspase-1 displayed no affinity for LPS (revised Supplementary Fig. 6), demonstrating that the caspy2 N-terminal pyrin domain is crucial for LPS recognition in zebrafish (revised Supplementary Fig. 6). However, both caspy and caspy2 were not precipitated by biotinylated-Pam3CSK4, another immunostimulatory peptide from bacterial lipoprotein (revised Fig. 3a).

Fig. 3c. One cannot compare an intact cell to a pyroptotic cell and make these conclusions. It appears that most of the green signal has been lost from the cell, consistent with pyroptosis lysing the cell and releasing the protein. The authors need to repeat this under conditions where there is no pyroptosis. I suggest doing the experiment in 10mM glycine to inhibit pyroptosis, as well as trying it in the presence of a caspase-inhibitor. DIC/phase images should be shown alongside to show lack of pyroptosis.

Reply: According to the reviewer's comments, we repeated the experiments and also used the cytoprotectant glycine to inhibit cell lysis during 0909I *E. piscicida* infection, and observed more caspy2 foci in wild-type ZF4 cells (Fig. 3c). The DIC/phase images were also added in the revised manuscript (revised Fig. 3c).

The methods mention purified Caspy2, but I did not see methods on how this was purified, simply an HA tag, or immunoprecipitated as suggested by Supp Fig. 4? Figure 3 legend states that Caspy2 proteins were "prepared" – what does that mean? Methods section says that the proteins were purified but does not say how this was done. Methods must be included.

Reply: Thank you for the constructive suggestion. We have corrected the processes in revised Methods, as well as in revised Figure legends.

Fig. 5a and 5b. mRNA says there is a 30-fold induction of Caspy2 expression in Fig. 5a. But I see almost no difference in total Caspy2 protein in Fig. 5b. How do the authors reconcile this discrepancy?

Reply: The reviewer raised a valid question. As we all known, the mRNA level does not always predict protein expression level, especially in an *in vivo* infection model, which was more complicated with both non-myeloid cells and myeloid cells. However, accordingly, we have repeated the experiments with a more specific rabbit anti-caspy2 antibody, and revised the panel in new Figure 6b, we assume that there's no difference in total caspy2 protein might have been because of the activation of this processes, and that the protein level varies dynamically to maintain homeostasis during development, as well as during infection.

The processing in Fig. 5b is quite weak. If one is willing to accept weak band, then the WT bacterial infection also causes the appearance of new weak band at around 35 kD. Thus, the data is questionable for its relevance.

Reply: Thank you for the constructive suggestion. Accordingly, we wondered the previous week band might be a non-specific signal. We tried our best to generate a more specific rabbit anti-caspy2 antibody to repeat the experiment. Compared with the activation of caspy2 during 0909I *E. piscicida* infection in zebrafish larvae, we could not detect the activation in EIB202 *E. piscicida* infection model (revised Fig. 6b).

You cannot say that the “control MO larvae were unaffected (Fig. 5c)” when one of them died and all had bacterial burdens in Fig. 5d.

Reply: We have corrected this in the revised manuscript.

Figure 5d is plotted inappropriately. There are dead fish in this experiment, the numbers of which are different starting at the earliest time point. Were the CFU burdens from dead fish included in the graph? If so, these carry the caveat that the fish was dead and bacteria will be replicating with no check from the fish immune system. If they were excluded, then statistical analysis cannot be performed because the highest points in the graph were excluded (assuming fish with highest burdens were the ones that died, which is likely). In either case, the dead fish need to be graphically represented on the graph. One could mark these with an X through the symbol, or instead of the symbol. If dead fish are excluded, X could be added above the axis of the graph to show that a data point is missing. In either case, statistical analysis will be difficult to perform.

Reply: The reviewer raised a valid question. We checked the bacterial burdens in both survived and dead larvae, but because the dead fish carry higher bacteria counts (See Fig. R2 below), it is very difficult to compare the bacterial burdens between the two groups. In our manuscript, we performed the experiments with large numbers of larvae. The surviving larvae at the indicated post-infection time points were collected, and homogenates were made for CFU counts (revised Fig. 6d). Each symbol represents the average counts of 5 larvae (n=100 for control-MO, n=100 for caspy2-MO).

Our data represents the bacterial loads in surviving animals, which were all checked by the immune system. We have corrected the panel and revised the legends accordingly.

Fig. R2. Dead zebrafish larvae were collected at the indicated post-infection time points.

Fig. 5c vs. Fig. 5i shows that the Caspy2 MO knocked down fish have greater mortality when infected with a low dose of 10^5 than they do at a high dose of 10^9 . This is biologically paradoxical, and leads one to wonder about the physiologic relevance of the infectious model.

Reply: We apologize for the confusion. According to previous studies, caspase-11 activation was initially found to cause sepsis in mouse via administration of LPS or pathogens infection (Hargar et al., *Science*. 2013, 110: 1250-1253; Kayagaki et al., 2013, 110: 1246-1249; Broz et al., *Nature*. 2012, 490: 288-291), and was subsequently found to promote mucosal defense against enteric bacterial pathogens *in vivo* (Knodler et al., *Cell Host & Microbe*. 2014, 16: 249-256; Pallett et al., *Mucosal Immunol*. 2017, 10: 602-612; Sellin et al., *Cell Host & Microbe*. 2014, 16: 237-284). The former model is detrimental to the host, while the latter is protective for the host, which seems to represent a double-edged mechanism to maintain the balance between infection and anti-infection. In our manuscript, we used 10^5 CFU/ml 0909I *E. piscicida* to infect both control- and *caspy2*-MO larvae, this low dose might trigger the *caspy2* activation *in vivo* at a level required for bacterial clearance (revised Fig 6b). However, when we treated the larvae with higher dose of 0909I *E. piscicida* (10^9 CFU/mL), which could trigger a hyper-activation of non-canonical inflammasome and result in septic pathophysiological effects. Both sepsis-level and antibacterial-level activation of non-canonical inflammasome were attenuated in *caspy2*-MO larvae (revised Supplementary Fig. 10). Thus, our results indicate that the *caspy2* activation-mediated pyroptosis might control anti-bacterial defense and lethal effects of non-canonical inflammasome activation in zebrafish.

Minor aspects.

The introduction should mention how many caspase genes are present in the zebrafish genome.

Reply: According to the reviewer's constructive suggestion. We have revised the Introduction.

"...and the formation of the p20 cleaved form of Caspy2 was detected in wild-type LPS-transfected ZF4 cells, but was impaired in caspy2-KO cells (Fig. 3e)." This is incorrectly stated. Of course the Caspy2 p20 is absent from knockout cells because it was knocked out, not that presence of the p20 was "impaired". Also, the p20 is so weak as to lead one to wonder whether cleavage has much meaning for Caspy2. Cleavage of caspase-11 has been a poor marker for its catalytic activity.

Reply: Thank you for the constructive suggestion. We have corrected the text in the revised manuscript.

Reviewer #2 (Remarks to the Author):

Review of Yang et al. for Nature Communications

Summary:

In this manuscript, Yang and colleagues examined the role of the zebrafish caspase family member Caspy2. Based on the homology of Caspy2 to mammalian Caspase 11/5 and its ability to cleave canonical Caspase 11 substrate, authors conclude that Caspy2 might represent a conserved zebrafish equivalent of non-canonical inflammasome in zebrafish. They provide data that suggest that Caspy2 has the ability to interact with LPS and suggest that this involves its PYD-domain. Furthermore, the data show that Caspy2 can somehow initiate pyroptotic-like cell death both in human HeLa cells and in zebrafish fibroblast ZF4 cell line, yet the mechanism remains unclear. Finally, the authors also perform in vivo study of Caspy2 expression and activation, involving Caspy-2-MO larvae, indicating a role for Caspy-2 in protecting fish larvae from bacterial infection in the gut. Moreover, they also establish and provide important data on Caspy2 function in a novel model of LPS-induced mortality in larvae.

Although homology of Caspy2 to mammalian Caspase 5/11 and its ability to cleave a Caspase 11/4/5 substrate has been demonstrated in several studies before, it was so far not shown whether Caspy2 is indeed a functional homolog of Caspase4/5/11 functions in zebrafish model. The study also provides intriguing observation that PYD domain of Caspy2 might be important for its LPS binding and activation, which, considering previously demonstrated LPS-binding activity of mammalian CARD-containing Caspase 11, suggests that this function might be conserved for the different death fold superfamily domains, and that PYD and CARD domain functions might have

been swapped during evolution of vertebrate immune system. However, more experiments are necessary to fully support this claim and to further characterize caspy-2 expression/induction and the Zebrafish non-canonical pathway. From an experimental aspect, the quality and presentation of the data needs improvement, and essential controls are still lacking (see below). In addition, the Figure legends and materials and methods section are poorly written and in many occLast, but not least, I am worried whether sound scientific practises were applied when generating the data based on discrepancies in the full western blots presented in Suppl. Figure 10 (for details see below).

General remarks:

1) The authors do not at all explain their choice of the bacterial strains used in the study, and the genetic difference between EIB202 and 0909I strains and do not provide evidence that this strain specifically activates only "non-canonical inflammasome" but not canonical pathway in zebrafish cells. How are these bacteria entering the cytosol of ZF4 cells upon infection, and how is this related the the genetic difference between EIB202 and 0909I strains. They do not provide a data indicating whether the canonical inflammasome pathway components are expressed or not in ZF4 cells, therefore it is difficult to assess potential canonical inflammasome contribution to phenotypes observed both in vitro and in vivo. It is also known from the literature that non-canonical inflammasome activation can lead to NLRP3 inflammasome activation in mammalian cells. Is this also a case in Zebrafish cells?

Reply: The reviewer raised a valid question. 0909I *E. piscicida* was screened from our transposon insertion mutant library, which was generated by conjugation between EIB202 *E. piscicida* (recipient) and SM10 λ pir/pMar2xT7 (transposon donor) (Yang et al., mBio, 2017, 8: 5e01581-17). From our unpublished data, we have demonstrated that 0909I is the hemolysin (ethA) overexpression strain. Compared with wild type *E. piscicida* infection, 0909I infection significantly promoted the caspase-4 dependent pyroptosis and IL-18 secretion in intestinal epithelial cells (IECs), as well as in HeLa cells (Supplementary Fig. 1). According to these evidences, we found that the overexpressed-hemolysin internalizes into cells via binding to OMVs and promotes rupture of OMV-containing vesicles, thereby releasing OMV-derived LPS into the cytoplasm and eventually triggering significant activation of non-canonical inflammasome. The relevant data have been submitted as a separate manuscript in parallel. If required, we can send the manuscript to the reviewer for reference.

In this work, based on the results in mammalian cells, we just take the advantage of 0909I *E. piscicida* as a tool to infect zebrafish fibroblasts cells, trying to find whether the non-canonical inflammasome activation exists in lower vertebrates and further to find which protein is the LPS-sensing caspase. And our results do show that 0909I *E. piscicida* infection, but not EIB202 *E.*

piscicida, triggers significant Caspy2 activation in Zebrafish fibroblasts (revised Fig. 2), which was responsible for pyroptosis in zebrafish non-myeloid cells (revised Fig. 2).

In relation to the canonical inflammasome activation in zebrafish cells, the reviewer raised a valid question. However, in zebrafish, although the key components of the inflammasome complex assembly are conserved, the nod-like receptors are less well understood (Li et al., Cell. Mol. Immunol. 2017, 14: 80-89). Moreover, in our study, when we treated ZF4 cells with the specific caspase-1 inhibitor, 09091 *E. piscicida* infection triggered cell death cannot be reduced (revised Fig. 1). Thus, we expect that 09091 *E. piscicida* infection could significantly induce the non-canonical inflammasome-gated pyroptosis (revised Fig. 2). Although the mechanism about whether this caspy2 activation can lead to “NLRP3 inflammasome” activation in zebrafish remains unknown, we feel that understanding of the mechanism involved is not required for the conclusions of this manuscript and would like to subject for future investigation.

2) The authors claim that Caspy2 is strongly upregulated upon bacterial infection (see Fig. 2), and, later, by LPS treatment (Fig. 3e). However, this data is conflicting with the *in vivo* data presented later Figures, where Caspy2 mRNA seems to be present constitutively in gills and gut (Fig. 4), and where Caspy-2 is expressed even in uninfected conditions (0h p.i., Figure 5b) and expression doesn't increase with time (Fig. 5b). The authors do not provide any explanation to this. Furthermore, in Figure 3e the authors present blots of cells that were apparently primed with LPS for 4 and then differentially treated. Here Caspy-2 expression is only observed when either LPS is used or LPS-CTB. This is confusing, since either long priming is necessary (4h +), or LPS needs to reach the cytosol to prime Caspy-2 expression. Therefore, it will be necessary to clarify these discrepancies and define under which conditions Caspy-2 is induced and what pathway drives its expression.

Reply: Thanks for the reviewer's constructive suggestion. As we known, ZF4 is a zebrafish embryos fibroblast like cell line, which is a non-meyloid cell. In our study, we have tested the expression of caspy2 in ZF4 cells in Mock, Pam3CSK4 priming, or infection conditions, and repeated the experiments with a more specific rabbit anti-caspy2 antibody, the expression of caspy2 is induced by infection in ZF4 cells (revised Fig 2 and 4). However, when caspy2 is expressed *in vivo*, the mRNA level does not always predict the protein level, especially in an *in vivo* infection model, which is more complicated, with both non-myeloid cells and myeloid cells. Accordingly, we have repeated the experiments, and revised the panel in new Figure 6b, we assumed that the lack of difference in the caspy2 protein *in vivo* might due to the activation of this processes, and the protein expression varies dynamically to maintain the homeostasis during development, as well as infection. The immune responses are constantly invoked, which might be more complicated than in *in vitro* non-myeloid cells.

For previous Figure 3e, we apologize for the confusion. In our previous data, we only primed the CTB+LPS group with LPS, because we cannot use LPS to prime and then stimulate with LPS because this might induce rapid cell death. That is why caspy2 expression was only observed when either LPS or CTB plus LPS were used. According to the reviewers' constructive suggestions, and according to previous studies (Yang et al. *Immunity* 2015, 43, 923–932; Hargar et al. *Science*, 2013, 341, 1250-3), we have repeated the experiment by priming ZF4 cells with Pam3CSK4 for 4 hours, then stimulating with LPS, CTB+LPS, or CTB, which was described in both revised Figure legends and the Methods. We found that Caspy2 expression is induced in ZF4 cells (Fig. 4c). Moreover, when we administrated CTB+LPS to the Pam3CSK4-primed ZF4 cells, a comparative activation of caspy2 was observed (Fig. 4c). Most importantly, this intracellular LPS-induced cell death is dependent on caspy2 (Fig. 4b and c).

3) The Western blot image in the Figure 2b shows the processing of Caspy2, with several minor bands appearing after infection. However, the authors do not provide any data indicating whether this processing is necessary for Caspy2 activation and pyroptosis, although they indicate it on the scheme (Supplementary figure 6). At the moment, it is disputed if processing is necessary for activation of mouse Caspase-11. Thus, the authors need to identify autoproteolysis sites, and confirm that autoprocessing is an essential prerequisite for caspy2 activity (pyroptosis, WEHD activity) by mutating potential cleavage sites.

Reply: According to the constructive suggestions from the reviewer. We have repeated the experiments with a more specific rabbit anti-caspy2 antibody, and included the EIB202 *E. piscicida* infection samples. Actually, caspy2 was not constantly expressed in zebrafish fibroblasts. Upon infection with *E. piscicida*, caspy2 expression was significantly induced, and the activation was detected in 09091-infected ZF4 cells (revised Fig. 2b). In addition, in revised Fig. 4e, when we mutated caspy2 at the catalytic activity site, the capsase-5-like activity (WEHD) was significantly blocked (Supplementary Fig. 4b), and the cell death was reduced (revised Fig. 4d and e). Thus, we expect that the catalytic activity site is required for caspy2 activation-induced pyroptosis in zebrafish.

4) What are the components of the noncanonical inflammasome in Zebrafish? In mice Caspase-11 only induces cell death via GSDMD, but can't process IL-1b. IL-1b processing is however observed after Casp-11 activation due to the engagement of the NLRP3-ASC-Casp-1 inflammasome that responds to GSDMD induced potassium release. Does Caspy-2 process Zebrafish IL-1b, or only Caspy-1. IS IL-1b production observed during infection with *E. piscicida*, and if yes, is it Caspy-2 or Caspy-1 dependent. Following these leads it should also be possible to identify the NLRP3 equivalent in Zebrafish.

Reply: The reviewer raised a valid question. Per the components of noncanonical inflammasome in zebrafish, we are trying to set up the CRISPR/cas9 library to screen the key components using the 0909I *E. piscicida*-infection model, and this project is still ongoing. For the IL-1 β procession in zebrafish cells, previous study has proved that both Caspy and Caspy2 can trigger the IL-1 β cleavage (Vojtech et al., *Infection and Immunity*, 2012, 80(8): 2878-85), which was quite different from mammalian cells. However, the mechanism about how zebrafish caspy2 activation induced IL-1 β production remain unknown, and whether this process is correlate with the unknown GSDMD mediated pore-formation as in mammals (Broz et al., *Euro. J. Immunol.* 2018) also needs to be clarified. Although this is important, we feel that that understanding of the mechanism involved is not required for the conclusions of the manuscript, but will be the subject of future investigations.

5) The data on Caspy-2 PYD binding LPS are too preliminary and require additional experiments. The authors need to show that the Caspy-2-PYD itself has the ability to interact with LPS, and ideally that transferring the Caspy-2 PYD on Caspy-1 transfers the ability to bind LPS (similar experiments were done by Feng Shao in Shi et al. *Nature* 2014). Furthermore, it needs to be shown that Caspy-1 does not bind to LPS nor other PYD-containing proteins (e.g. ASC). Additional ligands need to be tested as well, as done by Shi et al., such as biotin-Pam3CSK4 or MDP.

Reply: Thank you for the constructive suggestions. Accordingly, we repeated the pulldown assays to test the LPS-binding affinity and specificity of caspy2 and caspy in zebrafish (revised Fig. 3). Biotinylated-ultrapure LPS efficiently precipitated caspy2, caspy2 (C296A), and caspy2 PYD, but not caspy2 (Δ PYD), caspy, or caspy PYD (Fig. 3a). In addition, we used the biotinylated-Pam3CSK4 as the control, and neither caspy nor caspy2 could be detected from *in vitro* pulldown assays.

In addition, in enzyme linked immunosorbent assays (ELISAs) of LPS-binding activity, we added the controls according to reviewers' comments. Consistent with caspase-11, caspy2, caspy2 (C296A), and caspy2 PYD showed strong affinity to immobilized LPS, with the number of LPS-caspy2 complexes increasing in a concentration-dependent manner (revised Supplementary Fig. 6). However, caspy2 (Δ PYD), caspy, caspy PYD, and caspase-1 displayed no affinity for LPS (revised Supplementary Fig. 6), demonstrating that the caspy2 N-terminal pyrin domain is crucial for LPS recognition in zebrafish (revised Supplementary Fig. 6). However, both caspy and caspy2 were not precipitated by biotinylated-Pam3CSK4, another immunostimulatory peptide from bacterial lipoprotein (revised Fig. 3a).

6) The authors propose that oligomerization of Caspy-2 is induced by LPS binding and is required for Caspy-2 activation and auto-processing. Yet, no evidence is shown, except in Figure 3c, where somekind of Caspy-2 speckles can be observed. Oligomerization of Caspy-2 upon LPS binding should be assessed ideally both with purified protein, by performing size-exclusion analysis of

Caspy-2 with and without addition of LPS, and of the endogenous Caspy-2, by detecting the oligomeric form. The punctate structures in Fig. 3c alone do not provide an evidence of direct interaction between single Caspy2 molecules. It also cannot be ruled out that the punctate cytoplasmic structures that authors observe arise as a treatment artefact or, alternatively, contain additional partners involved in LPS binding and pyroptosis initiation. Authors also claim that there was a significant difference in number of these structures observed upon infection with two strains, however, this data is not included to the manuscript (quantification).

Reply: We agree with the reviewer's suggestions. Accordingly, we repeated and revised the experiments. For *in vitro* oligomerization of caspy2, consistent with the binding data (Fig. 4a), we treated the samples with 0.5% Triton X-100 to reduce the LPS-induced Caspases into a smaller homogenous form on blue native PAGE (Shi et al., Nature, 2014, 514: 187–192). We found that purified HA-tagged caspy2 and caspy2 PYD from transfected HEK293T cells were oligomerized upon incubation with LPS, but not with Pam3CSK4 (revised Fig. 3b).

to detect endogenous caspy2 oligomers, wild-type or *caspy2*-KO ZF4 cells were stimulated with 0909I or EIB202 *E. piscicida*, and digitonin-solublized cell lysates were resolved by blue native PAGE with 0.5% Triton X-100, and then western blotting was performed using anti-caspy2 antibodies. A large oligomeric complex contain caspy2 was induced in wild-type ZF4 cells upon infection with 0909I, but not with EIB202 *E. piscicida* (revised Fig. 3d).

For quantification of caspy2 foci during infection, we repeated the experiments accordingly and added the results to revised Fig. 3c.

7) Authors do not provide enough data on Caspy2 role in their Zebrafish larvae septic shock model. They do not state what physiological and morphological changes occur in larvae, and whether Caspy2 activation leads to the cytokine processing and release. Therefore, the relevance of this model to existing septic shock models in higher vertebrates remains unclear.

Reply: The reviewer raised a valid question. Accordingly, we have added the lethal sepsis physiological and morphological changes occurring in larvae according to previous studies (Barber, 2016, University of Utah, ProQuest: 10163521; Na et al., PLoS ONE, 2015, 10, e0118203), and revised the manuscript and Figures (Results and Supplementary Fig.10).

8) The western blots shown in Figure S10 are a major point of criticism. A) In general it has the appearance that the authors just cut out what they liked from these blots, no information is for example provided on what the other bands are from these blots. B) More seriously, I am quite shocked by the fact that the b-actin controls do not appear to arise from the same blot as the Caspy-2 blots for example. Fig. 3e provides one example where Caspy-2 data were obtained from 2 blots, while a 3rd blots with some random bands apparently serves as the b-actin control. To convince me

that equal amounts were loaded, the authors should strip the Caspy-2 blots and re-blot for b-actin. C) Most serious: The blot shown for Fig. S9a shows the timecourse of Caspy-2 induction in normal larvae (lane 1-6) and Caspy-2-MO larvae (lanes 7-12). The crossreactive band at high molecular weight shows can serve to identify the different lanes (i.e. 0.5, 1, 3 and 5 dpf). Shockingly the blot suggests that the authors selected lanes 2-5 to show induction of Caspy-2 in Wt larvae, while showing lanes 7-10 in the Caspy-2 MO larvae. Based on the upper band, I think that lanes 8-11 should have been selected, which would show that there is no difference in Caspy-2 induction between the WT and presumed Caspy2-MO's, which would question the validity of all in vivo data in the study! Furthermore, the blot suggests that the authors intentionally do not show a later timepoint at which Caspy-2 is expressed in Caspy-2-MOs! Since the lanes are not labelled and no more explanation is given for the blots shown in S10, I would ask the authors to comment on my remarks. Also, b-actin blots in S9a cannot possibly be from the same membrane as the Caspy2 blots.

Reply: We are sorry for the confusion about the western blot data. According to the reviewers' comments, we have tried our best to repeat and carefully revised our Figures. As for the β -actin probing, in our laboratory, we always use the same amount of lysate to run the gel and probe with β -actin antibody (1:5000; Abcam) as a loading control (Supplementary Fig. 11). As for the previous Fig. S9a, we are sorry for making the confusion; we selected the 0.5-5 dpf larvae samples to show the MO knockdown efficiency, because we used the 5 dpf larvae for experiments. The previous lane 8-11 (in previous Fig. S9) were the 10 dpf larvae samples, where the caspy2 expression was recovered in caspy2-MO larvae, that's why there's no difference in protein induction. To better address the reviewer's concern, and make sure the phenotype is correct in our *in vivo* results, we checked more time points post fertilization in the MO treated groups, immunoblotting showed that this MO effectively knocked down caspy2 up to and including 7 dpf (Supplementary Fig. 9c), and confirmed that the control MO had no effect on caspy2 expression (Supplementary Fig. 9c). We also showed that the Caspy2 expression was recovered in caspy2-MO treated larvae at 10 dpf; however, the *in vivo* experiments in our manuscript were performed for no longer than 10 dpf. Thus, we believe that our data is valid.

Additional comments:

In the introduction, authors should elaborate more on general importance of inflammasome pathway in vertebrates, as well as advantages of Zebrafish as a model to study innate immune response. The objectives of the study need to be stated more clearly.

Reply: According to the reviewer's constructive suggestion. We have revised the Introduction.

Fig 1: The authors observe high levels of cytotoxicity (up to 20%) in their controls in Figures 1,2,3. However, they do not provide an explanation to this phenomenon. For example, it is unclear how the LDH levels were calculated. Normally, % LDH release is normalized to uninfected or mock-infected samples. This does not appear to be the case, as in Fig. 1a, mock-infected samples show already 20% of LDH release. What was the zero-value that was used to calculate the percentage of LDH release?

Reply: Thank you for the constructive suggestion. Normally, we do the infection experiments by replacing complete medium with Opti-MEM, and culture for 2 hours, which resulted in 10-20% cytotoxicity as background in ZF4 cell. However, the conditions were same with EIB202 or 0909I *E. piscicida* infection, and the data were pooled from at least 3 independent experiments. However, this is quite normal compared with the reported cytotoxicity in other studies (Yang, D., et al. Immunity 2015, 43, 923–932; Hargar et al. Science, 2013. 341, 1250-3; Kayagaki, N., et al. Science 2013, 341, 1246-9). For the protocol about LDH release assays, we have corrected this in revised Methods. Accordingly, aliquots of cellular supernatants were transferred into 96-well plates (round bottom) and centrifuged at $1000 \times g$ for 5 min. The supernatants were transferred to another 96-well plate (flat bottom), and the plate was subjected to the cytotoxicity assay using a CytoTox 96 assay kit (G1780, Promega, Madison, WI, USA) according to the manufacturer's protocol. Each sample was tested in triplicate. Cytotoxicity was normalized to Triton X-100 treatment (100% of control), and LDH release from uninfected/untreated cells was used for background subtraction.

Fig. 2b: Confirm that CRISPR-KO of Caspy-2 does not change the activation of canonical inflammasomes, which could account partially for the cell death.

Reply: Per the canonical inflammasome activation in zebrafish cells, the reviewer raised a valid question. However, in zebrafish, although the key components of the inflammasome complex assembly are conserved, the nod-like receptors are less well understood (Li et al., Cell. Mol. Immunol. 2017, 14: 80-89). Moreover, in our study, when we treated ZF4 cells with the specific caspase-1 inhibitor, 0909I *E. piscicida* infection triggered cell death cannot be reduced (revised Fig. 1). Thus, we expect that 0909I *E. piscicida* infection induce the non-canonical inflammasome-gated pyroptosis is more important in our model (revised Fig. 2). Although the mechanism about whether this caspy2 activation can lead to “NLRP3 inflammasome” activation in zebrafish remains unknown, we feel that understanding of the mechanism involved is not required for the conclusions of this manuscript and would like to subject for future investigation.

Fig. 3a. Equal amounts of protein need to be loaded in the input, since like this it is impossible to judge if really no Caspy2-Dpyd has been pulled down.

Reply: We have repeated and corrected the panel in the revised manuscript.

Fig. 3a: There is no description how the protein was purified. If purified from bacterial sourced, we can expect it to bind LPS already, which would interfere with the LPS PD>

Reply: Thank you for the constructive suggestion. We have corrected the processes in the revised Methods, as well as in the revised Figure legends.

Fig. 3a: What is the second band in the 4th lane of the second blot that runs at 24 kDa?

Reply: Thank you for the constructive suggestion. Accordingly, we wondered whether the previous week band might be a non-specific signal. In our revised manuscript, we repeated the experiments carefully, and purified all the HA-tagged caspase proteins from the indicated plasmids-transfected HEK293T cells using Pierce Anti-HA magnetic beads, and used the rabbit anti-HA antibody to detect the indicated bands in revised Fig. 3a.

Fig. 3b: How was the ELISA used in this experiment validated. Data need to be shown that it works with other LPS-binding proteins and a negative control (e.g. Casp-1) needs to be added as well.

Reply: We agree with the reviewer's suggestions, and have added the controls to repeat the experiments. Consistent with caspase-11, caspy2, caspy2 (C296A), and caspy2 PYD showed strong affinity to immobilized LPS, with the number of LPS-Caspy2 complexes increasing in a concentration-dependent manner (revised Supplementary Fig. 6). However, caspy2 (Δ PYD), caspy, caspy PYD, and caspase-1 displayed no affinity for LPS (revised Supplementary Fig. 6), demonstrating that the caspy2 N-terminal pyrin domain is crucial for LPS recognition in zebrafish (revised Supplementary Fig. 6). However, neither caspy nor caspy2 could be precipitated by biotinylated-Pam3CSK4 (revised Fig. 3a; revised Supplementary Fig. 6).

Fig. 3c: I am intrigued by the Caspy-2 speckles. No comparable structures can be seen for mouse Caspase-11 during LPS TF or bacterial infections. Are these oligomers of Caspy-2, or Caspy-2 binding to the bacterial surface? Can these oligomers be detected by SDS-PAGE/WB after X-linking?

Reply: Thank you for the constructive suggestion. We repeated the experiments accordingly. Consistent with the caspy2 foci observed in 0909I *E. piscicida*-infected wild-type ZF4 cells, caspy2 treated with 4 mM DSS remained as oligomers on the SDS gel, which were triggered by infection with 0909I, but not with EIB202 *E. piscicida* (Fig. 3d). Consistently, we also performed the experiments to detect the endogenous caspy2 oligomers according to previous studies (Shi et al.,

Nature, 2014, 514: 187–192; He et al., Nature, 2016, 530: 354-357). Wild-type or *caspy2*-KO ZF4 cells were stimulated with 0909I or EIB202 *E. piscicida*, and digitonin-solubilized cell lysates were resolved by blue native PAGE with 0.5% Triton X-100, and the blots were then immunoblotted with anti-caspy2 antibodies. A large oligomeric complex contain caspy2 were induced in wild-type ZF4 cells upon infection with 0909I, but not with EIB202 *E. piscicida* (revised Fig. 3d). Thus, our results suggested that bacterial infection induced caspy2 oligomerization in zebrafish fibroblast.

Fig. 3e: This experiments needs to be repeated under conditions where cells have been primed to express Caspy-2 (Pam3CSK4 priming?). Based on what is shown here, we cannot exclude that CTB alone would activate Caspy-2, since no Caspy-2 is present in lane 3.

Reply: According to the reviewers' constructive suggestions and previous studies (Yang et al. Immunity 2015, 43, 923–932; Hargar et al. Science, 2013. 341, 1250-3), we have repeated the experiment by priming ZF4 cells with Pam3CSK4 for 4 hours, and then stimulate with LPS, CTB+LPS, or CTB, which was described in both the revised Figure legends and the Methods. We found that caspy2 expression is induced in ZF4 cells (revised Fig. 4c). Moreover, when we administrated the Pam3CSK4-primed ZF4 cells with CTB+LPS, a comparative activation of Caspy2 was observed (revised Fig. 4c), and most importantly, this intracellular LPS induced cell death is dependent on Caspy2 (revised Fig. 4b and c).

Fig. 3f: Additional controls necessary such as complementation of Casp4^{-/-} HeLa with Casp-4 and also Casp-11.

Reply: According to the reviewer's suggestion, we expressed the wild-type, C254S, or ΔCARD form of caspase-11 in CASP4^{-/-} HeLa cells. Consistently, wild-type caspase-11, but not the C254S or ΔCARD mutants, restored intracellular LPS-induced pyroptosis in these cells (Supplementary Fig. 7). These observations suggest that zebrafish caspy2 pyrin domain has a function similar to that of caspase-11 CARD domain in recognizing intracellular LPS (Supplementary Fig. 8), and that the catalytic activity is required for pyroptosis in zebrafish.

Fig. 3g: compared to WT CAAspy2, D296A and Dpyd are poorly expressed. Needs to be repeated with equal amounts.

Reply: We have repeated the experiments and revised the figure in the manuscript.

Fig. 3f/g: What is the contribution of human Casp5 in these assays?

Reply: In HeLa cells, caspase-5 expression was not enriched compared with other cells, (e.g. HHStC, CACO2, or hTCEpi cells (Human Protein Atlas)). Moreover, according to a previous study (Shi et al., Nature, 2014, 514: 187–192), the intracellular LPS induced pyroptosis was dependent on

Caspase-4 in HeLa cells, while in Caspase-4 KO HeLa cells, the cell death was abolished during non-canonical inflammasome activation. Based on the results, our manuscript used the same Caspase-4 KO HeLa cell line, which was generated in Feng Shao's lab, to reconstitute caspy2 or caspase-11, and detected the pyroptosis induced by intracellular LPS. Our data showed that both caspase-11 and Caspy2 are essential in intracellular LPS-triggered non-canonical inflammasome activation in HeLa cells (revised Fig. 3 and Supplementary Fig. 8).

Fig. 4b: What is the 42 kDa band that appears in lane 2-4. Is it a splice variant of Caspy-2? And why does it disappear with time? Another issues: markers seemed to be different in the full blots (Fig. S10).

Reply: Thank you for the constructive suggestion. Accordingly, we wondered the previous week band might be a non-specific signal. We generated a more specific rabbit anti-caspy2 antibody to repeat the experiment and revised the data in new Fig 5b.

Fig. 4c: there is a strong In-situ hybridisation signal for the embryo at 12 hpf, which can be viewed as a high level of Caspy2 mRNA. This is controversial to the data from the panels a and b of the same figure.

Reply: We are sorry for the confusion. According the reviewer's comments, we have repeated the experiments, and found that the signaling at 12 dpf was caused by the egg membrane, which would stain inside, and is difficult to wash out. We removed the egg membrane and repeated the experiments (revised Fig. 5c).

Fig 4: It is known from literature that mRNA level does not always predict protein expression level. Therefore, it could be interesting to check Caspy2 protein expression in embryos during different developmental stages, as well as at different time points before and after infection, using whole embryo immunofluorescence approach.

Reply: We agree with the reviewer's comments. Accordingly, we repeated and added the total Caspy2 protein levels through western blot analysis during zebrafish larvae development (revised Fig. 5b), and also checked the Caspy2 expression in embryos at different time points before and after *E. piscicida* infection (revised Fig. 6b).

Fig5a: What pathway is engaged that induces Caspy-2 in vivo?

Reply: The reviewer raised a valid question. In mammalian, caspase-11 was proved to directly recognize intracellular LPS, and trigger the GSDMD mediated pore-formation (Shi et al., Nature. 2014, 514: 187-192; Shi et al., Nature. 2015, 526: 660-665), however, the pathways engaged in these signaling still needs further clarification. In our laboratory, we are trying to generate the

CRISPR/cas9 library to screen the pathways engaged in zebrafish caspy2 activation through 0909I *E. piscicida* infection, which would be critical for better understanding the pathways engaged with caspy2 *in vivo*, but this is still ongoing. While in our current manuscript, we revealed the function of Caspy2 in gating non-canonical inflammasome activation both *in vitro* and *in vivo*, we feel that understanding of the pathway involved is not required for the conclusions of this manuscript and would be the subject of future investigation.

Fig.5b: Why does the infection with EIB202 also induce the processing of Caspy-2? Why is there no induction at protein level with time? And Why is Caspy-2 already made in lane (0 hpi)?

Reply: According to the reviewer's concern, we wondered the previous week band might be a non-specific signal. In our revised manuscript, we generated a more specific rabbit anti-Caspy2 antibody to repeat the experiment, compared with the activation of caspy2 during 0909I *E. piscicida* infection in zebrafish larvae, we could not detect the activation in EIB202 *E. piscicida* infection model (revised Fig. 6b). As for the caspy2 already made in 0 hpi, which was the 5 dpf larvae, as we can see from the caspy2 expression during zebrafish development, caspy2 should have expressed at this time point (revised Fig. 5b).

Fig. 5i: On what cells does LPS act and how does it enter the larvae? How does LPS reach the cytosol of cells *in vivo* to cause Caspy-2 activation?

Reply: The reviewer raises a valid point. But so far how LPS enters the cytosol of cells *in vivo* remains unclear. Previous studies showed that TLR4 was not required for activation of the mouse non-canonical inflammasome *in vivo* (Harggar et al. Science, 2013. 341, 1250-3; Kayagaki et al. Science 2013, 341, 1246-9). Many others have studied and published on the model using a single administration of LPS *in vivo* (Kayagaki et al. Science 2013, 341, 1246-9; Na et al., PLoS ONE, 2015, 10, e0118203). We do not feel that further analysis of the model will add much to the current work, which focuses on the noncanonical inflammasome pathway. We performed our experiments using the LPS-induced sepsis model according to previous studies (Barber, 2016, University of Utah, ProQuest: 10163521; Na et al., PLoS ONE, 2015, 10, e0118203), and provided evidence that caspy2 was responsible for the zebrafish model of lethal sepsis (revised Supplementary Fig. 10), offering the possibility to study the inflammation activation and pathogen responses simultaneously in a whole organism

Suppl. Fig. 4: I am confused by this figure. The authors claim that LPS is the ligand that activates Caspy-2, yet they observe its activation in cell extract of transfected cells, that did were neither infected nor LPS transfected.

Reply: According to the concerns of the reviewer. We performed the experiments according to a previous study (Masumoto et al., J. Biol. Chem. 2003, 278 (6): 4268-4278), which was the first identification of zebrafish Caspy and Caspy2. Accordingly, the caspases enzymatic site is correlated with their caspase-type-specific substrates activity. Our results also claimed that the enzymatic activity site is critical for WEHD activity. However, the mechanism between LPS binding and catalytic activity with pyroptosis require further clarification.

Suppl. Fig. 9a: Here again a band is seen on the blots (at ~40 kDa) that is not commented on at all. Intriguingly it is suppressed by Caspy-2 MO, thus unlikely to be a cross-reactive band.

Reply: Accordingly, we generated a more specific rabbit anti-caspy2 antibody to repeat the experiment and revised the data in new Supplementary Fig 9c.

Supplementary fig, 9a, wt control is missing.

Reply: Thank you for the constructive suggestion. We have shown the caspy2 expression in wild-type control larvae in revised Fig. 5b.

Page 8, lines 150-153: This needs reformulating, since the data suggest that Caspy-2 has a function similar to Casp4 rather than Casp5. Please provide the references that show that Casp4 or Casp5 CARD binds LPS. I am not aware of such papers.

Reply: Thank you for the suggestion. We have corrected the text in the revised manuscript.

We believe that we have addressed the comments raised by the reviewers. In our opinion, the new results and revisions made in response to the constructive criticisms by the reviewers have resulted in a greatly improved manuscript. We hope that the revised manuscript is now suitable for publication in Nature Communications.

Sincerely,

Qin Liu (on behalf of all authors)

Reviewers' comments:

Reviewer #1 (Remarks to the Author):

The authors present a greatly strengthened manuscript with the addition of some impressive new data that includes controls that address all my comments. They now show Caspy controls that are negative for LPS interaction while Caspy2 is activated by LPS. They show that Caspy2 is not activated/interacting with another lipid PAMP (PAM3CSK4). They have new in vivo data that shows that LPS sepsis is ameliorated in Caspy2 morpholino knockdown fish. Together these data have convinced me of the conclusions. The authors also explain that the mutant is a hemolysin overexpressing mutant. There is significant impact to the authors findings as they note, revealing that the evolution of the death domains of caspases is complex, and the function of LPS detection can be accomplished by a pyrin in zebrafish, but a CARD in mice and humans. This has quite important implications for the death domain family, expanding our scope of the possible functions of the domains.

Overall, the paper is quite close to being suitable for publication in Nature Communications. However, the paper is rather poorly proofread and need a good editing before it is suitable for publication. Also, the high dose infection data does not make sense, however if removed from the paper, I think the conclusions stand and the paper is publishable with Supp Fig. 10a-c removed. Additionally, I have one request for additional data which should be very simple experiments for the authors to perform.

Comments:

That the mutant bacteria are overexpressing hemolysin and that wild type bacteria are not detected shows that caspy2 has nothing to do with defense against wild type *Edwardsiella*. This should be made very clear in the discussion. I think that it is ok to use the mutant to discover that zebrafish have a cytosolic LPS sensor, but that the focus of the early parts of the paper should be more directed at LPS and less at the bacteria. To this end my one request for more data is that the authors should add LPS transection or CTB-LPS to Figure 1, showing that after ultrapure LPS delivery to the cytosol there is cytotoxicity, WEHD caspase activity, and z-WEHD-FMK inhibitor blocks this specifically. Essentially repeat Figure 1A, 1B, 1C, and 1D with transfected LPS and expand Figure 1 to include this data. This is a very simple series of experiments that should only take a week to perform.

In the introduction the authors discuss vaguely that there are several caspases, not saying how many there are. They could say there are "at least X caspases... including caspase-C, caspy and caspy2". And are they mixing their nomenclature since it seems that caspy2 is also called caspyB which would make these caspase-A, B, and C? clarity is needed.

Supp Fig. 3. The annotated pyrin domain of zebrafish caspy2 overlaps with the caspase protease domains within the human caspases. The human caspase-4 protease domain starts somewhere in the neighborhood of MEAGPPES..... a sequence which is directly overlapping with the proposed pyrin domain. When I blast caspy2 amino terminus (from aa 1-140), the pyrin domain is called as amino acids 5-84 in the "superfamily hits" on BLAST. It does not hit to any human or mouse pyrin domains by simple BLAST search. The authors need to re-visit their alignments. Please look at the annotation in reference 10 from Inohara's paper on caspy where they annotate the pyrin domain as amino acid 1 through ...RLESN.

At line 85 the authors jump from having caspase substrate cleavage data to seemingly assuming that this arises from caspy2 and not other caspases. They state that caspy2 is 57% similar to human caspase-5, but do not state whether this is significantly higher than its similarity to other human caspases, and also do not state whether caspy and caspase-C might also be similar to human caspase-5. So the rationale for focusing on caspy2 was not described, and seems like an

unjustified leap in logic.

Authors refer to EIB202 strain in the figure panels and in the text and never define this as "wild type" except in the methods section. This makes for a confusing read. This strain could simply be referred to as "wild type". Actually, the authors do sometimes refer to the strain as wild type in other areas of the paper.

The extent of results text on fig. 2c is: "Moreover, 09091 *E. piscicida* infection induced caspy2 expression and the cleavage of pro-caspy2, as determined by immunoblotting using an antibody that recognizes p20, the cleaved form of caspy2 (Fig. 2c)." The authors fail to distinguish between transcriptional induction of caspy2 protein expression, as compared to activation of the protein measured by p20 protein band processing. They also do not mention that WT bacteria are also on the blot. The blot shows that transcriptional induction is driven by both WT and mutant bacteria, but processing only by the mutants. This actually bolsters the authors case. This oversight and several others give the impression that the authors have not put a lot of time into crafting the text of this manuscript.

"Taken together, these findings indicated that the existence of pyroptosis in zebrafish is mediated by caspy2 activation during bacterial infection." Given that the authors show that WT bacteria do not cause pyroptosis, this sentence is actually incorrect.

Human caspase-4/5 and mouse caspase-11 have a CARD... "However, the N-terminal domain of caspy2 is more homologous to the pyrin death domain (ref 10) (Supplementary Fig. 3)." Supp Fig. 3 does not show what this sentence says it shows. That information is in reference 10, but Supp Fig. 3 only aligns caspy2 to caspases, not to any pryin domains.

Fig. 4d and 4e are called figure 3d and 3e in the text. Please check figure callouts.

Fig. 6a and 6b contradict each other. 6a shows that WT bacteria do not induce expression of caspy2, while mutant bacteria do induce expression. Yet 6b shows that total caspy2 protein levels are actually somewhat higher in the WT bacteria. This data need to be clarified or removed from the paper. I think the western blot is more definitive and important as it shows caspy2 processing, and it is consistent with earlier data in Fig. 2c where both WT and mutant bacteria induce expression of caspy2 pro-protein. That said, exclusion of dead fish (see below) could be problematic in 6b as well.

Figure 6D. The authors have clarified in the figure legend that dead fish were excluded. However, these still need to be shown on the graph - the reader needs to be able to see how many dead fish were in the experiment. So, assuming that at the 72 hour time point 60% of the fish were dead, there should be something like 6 symbols of "X" shown together at the top of the graph (perhaps above the top of the Y axis) to represent dead and uncountable fish, and there should be 4 symbols of "O" showing the counts of the remaining live fish. However, I have the impression that the authors did not keep track of how many dead fish there were in the experiment, and that these were simply discarded without counting. I have this impression because at 72 hours it seems that there are equal numbers of dots between the Caspy2 MO and the Control MO. However, 60% of the former should be dead whereas only 10% of the Control MO should be dead. This suggests that the authors increased their sampling of the Caspy2 MO fish to obtain more live fish. This is not appropriate because it includes selection bias against the dead fish. Such a method would invalidate the time points where numbers of dead fish were very different from each other, so the 48 and 72 hour time points are unpublishable. If the authors have the numbers of dead fish from these experiments, and can include them on the graph, that is acceptable. If such data is not available, then the 48 and 72 hour time points should be removed from Figure 6D. This also affects Figure 6A and 6E and thus the 48h and 72h time points should be removed. None of this significantly affect the authors conclusions, it is simply not scientifically correct to present data from time points where dead animals are excluded.

Please remove Supp Figure 10a-10c because the data needs to be further investigated before it is published. By comparing Fig. 6c we can see that Caspy2 MO fish only 40% survive when infected with 10^5 CFU. One would expect that if the dose were increased, more fish should die. Yet at the 10^9 dose in Supp Fig. 10a-c, fewer fish die, with about 65% surviving. This makes no sense. If one does not compare Fig. 6c to Supp 10a, then each figure alone makes sense: 6c shows us that caspy2 promotes resistance to infection; 10a shows us that caspy2 is detrimental in sepsis. However, together they make no sense as a 10,000 fold increase in the dose should not cause more fish to survive. We must hypothesize that the obscenely high dose caused a secondary defense pathway to be activated, and this pathway exerted a dominant effect over the defect in caspy2 and that this secondary pathway rescued the fish from death. That needs to be worked out before the authors publish Supp Fig. 10a-c. That can be done in future work, and thus I recommend removing Supp Fig. 10a-c from this paper.

To me, the authors do not need to identify the gasdermin effector of pyroptosis that is downstream of caspy2 in this paper. I think the point that caspy2 is the LPS sensor can be made without that knowledge. That could be done in a subsequent study.

Reviewer #2 (Remarks to the Author):

The manuscript has clearly improved a lot and I congratulate the authors on the nice LPS interaction studies. However several of my original point have not yet been fully addressed.

Point 2)

What do the authors mean by:

"In our previous data, we only primed the CTB+LPS group with LPS, because we cannot use LPS to prime and then stimulate with LPS because this might induce rapid cell death. That is why caspy2 expression was only observed when either LPS or CTB plus LPS were used."

What rapid cell death is this? LPS alone should not induce cell death, or does it? And if yes, via Caspy-2?

Point 3) appearance of a p20 band of Caspy-2 after infection (Figure 2c)

I believe that the catalytic site is necessary for Caspy-2 activity, but this has nothing to do with my question.

My question is: What is the p20 band, and is generation of the p20 necessary for activity.

Point 4) Components of the non-canonical inflammasome in Zebrafish:

To justify publication in a high impact journal like Nature Communication, I believe it is not sufficient to only show that Caspy2 binds LPS i.e. is the Zebrafish equivalent of human Caspase-4/-5, mouse Caspase-11, but the whole non-canonical pathway needs to be analyzed in detail. After all, the first papers to show that showed that LPS activates human caspase-4/-5 were published 'only' in European Journal of Immunology (Schmid-Burgk JL EJI 2015 and Baker PJ EJI 2015) despite the higher relevance of the human system.

I understand that it might be difficult to identify the Zebrafish Nlrp3 homolog, given the large number of NLRs in the fish, but I would encourage the authors to at least check 1) if zASC specks are formed after caspy-2 activation, if 2) Caspy-1 is processed after LPS transfection and if 3) this is ASC dependent, 4) if mature zIL-1b is released after caspy-2 activation and 5) if Caspy-2 processes zIL-1b directly or via Caspy-1.

These experiments should highlight similarities and differences between the human/mouse and zebrafish system, and are relevant for the in vivo infection as well as the septic shock models, since currently it is not clear which pathways downstream of caspy-2 drive the response.

Point 5) interaction of LPS-Caspy2-PYD

The authors provide very nice proof of interaction of LPS with the PYD. A last experiment would be to replace the Casp-11 CARD with the PYD from Caspy2. Would this hybrid caspase be able to sense LPS and activate?

Point 6) Oligomerization of Caspy2.

I find the data about oligomerization of Caspy2 more convincing than before. However, they still do not address the question whether this oligomerization induces Caspy2 processing. For example, in Fig. 3d, there is a minor band (probably Caspy2 p20) in 0909I infected cell lysate. However, it is not visible upon LPS addition and oligomerization in Fig. 3b. It might indicate that additional players are necessary to induce Caspy 2 processing. Also, I was surprised to see that this band is not visible in Glycine-treated cells, and Caspy2 oligomers run lower in this sample.

One particular point that would be interesting to check is whether zASC is involved oligomerization of Caspy-2. In the mouse/human system non-canonical inflammasome activation does not require ASC. However in Zebrafish the ASC speck work differently, in that both ASC speck formation and caspase activation are induced via the PYD of ASC. Thus it might be possible that the PYD of Zebrafish ASC participates in the oligomerization of the caspase after LPS sensing.

Fig 2b: This does not address my issue.

The fact that a caspase-1 inhibitor doesn't block cell death in ZF4 cells doesn't mean anything unless it can be shown that it block activation of Casp-1 after stimulation of the canonical inflammasome pathway.

Fig 3b:

Since this assay might prove to be very useful to other researchers, I would suggest to also include a schematic of the assay

Suppl. Fig 4: The response doesn't not answer my question.

Why is no LPS needed to detect caspase activity in this setting?

We would like to thank the editor and the reviewers for the comments and insightful suggestions on our revised manuscript. Based on these comments, we have performed additional experiments and clarified certain statements in our revised manuscript. Our point-to-point response to the reviewers is provided below:

Reviewers' comments:

Reviewer #1 (Remarks to the Author):

The authors present a greatly strengthened manuscript with the addition of some impressive new data that includes controls that address all my comments. They now show Caspy controls that are negative for LPS interaction while Caspy2 is activated by LPS. They show that Caspy2 is not activated/interacting with another lipid PAMP (PAM3CSK4). They have new in vivo data that shows that LPS sepsis is ameliorated in Caspy2 morpholino knockdown fish. Together these data have convinced me of the conclusions. The authors also explain that the mutant is a hemolysin overexpressing mutant. There is significant impact to the authors findings as they note, revealing that the evolution of the death domains of caspases is complex, and the function of LPS detection can be accomplished by a pyrin in zebrafish, but a CARD in mice and humans. This has quite important implications for the death domain family, expanding our scope of the possible functions of the domains.

Overall, the paper is quite close to being suitable for publication in Nature Communications. However, the paper is rather poorly proofread and need a good editing before it is suitable for publication. Also, the high dose infection data does not make sense, however if removed from the paper, I think the conclusions stand and the paper is publishable with Supp Fig. 10a-c removed. Additionally, I have one request for additional data which should be very simple experiments for the authors to perform.

Comments:

That the mutant bacteria are overexpressing hemolysin and that wild type bacteria are not detected shows that caspy2 has nothing to do with defense against wild type *Edwardsiella*. This should be made very clear in the discussion. I think that it is ok to use the mutant to discover that zebrafish have a cytosolic LPS sensor, but that the focus of the early parts of the paper should be more directed at LPS and less at the bacteria. To this end my one request for more data is that the authors should add LPS transection or CTB-LPS to Figure 1, showing that after ultrapure LPS

delivery to the cytosol there is cytotoxicity, WEHD caspase activity, and z-WEHD-FMK inhibitor blocks this specifically. Essentially repeat Figure 1A, 1B, 1C, and 1D with transfected LPS and expand Figure 1 to include this data. This is a very simple series of experiments that should only take a week to perform.

Reply: The reviewer raised a valid question. Firstly, we have revised the Discussion accordingly, which were highlighted in the text. Secondly, we have performed additional experiments for Figure 1 as the reviewer requested. Consistent with 0909I *E. piscicida* infection, ZF4 cells exhibited significant cytotoxicity (Fig. 1e). The cytotoxicity induced by cytosolic LPS delivery was effectively inhibited by Z-WEHD-FMK and Z-VAD-FMK, but not Z-YVAD-FMK, Ac-DEVD-CHO, and Ac-LEVD-CHO (Fig. 1f). Moreover, lysates of LPS-transfected ZF4 cells showed significant preferential cleavage of Ac-WEHD-AFC, but did not cleave the other caspase substrates tested (Fig. 1g). Taken together, we believe that our results would be more directed at cytosolic LPS-induced non-canonical inflammasome is present in zebrafish.

In the introduction the authors discuss vaguely that there are several caspases, not saying how many there are. They could say there are "at least X caspases... including caspase-C, caspy and caspy2". And are they mixing their nomenclature since it seems that caspy2 is also called caspyB which would make these caspase-A, B, and C? clarity is needed.

Reply: We have revised the Introduction accordingly (page 3, line 37-44).

Supp Fig. 3. The annotated pyrin domain of zebrafish caspy2 overlaps with the caspase protease domains within the human caspases. The human caspase-4 protease domain starts somewhere in the neighborhood of MEAGPPES..... a sequence which is directly overlapping with the proposed pyrin domain. When I blast caspy2 amino terminus (from aa 1-140), the pyrin domain is called as amino acids 5-84 in the "superfamily hits" on BLAST. It does not hit to any human or mouse pyrin domains by simple BLAST search. The authors need to re-visit their alignments. Please look at the annotation in reference 10 from Inohara's paper on caspy where they annotate the pyrin domain as amino acid 1 through ...RLESN.

Reply: Thank you for the constructive suggestions. We have revisited the Inohara's paper, and we found that they have already revealed the N-terminal domain of caspy2 is most homologous to that of human Cryopyrin/PYPAF1 (46% similarity), a pyrin domain containing Nod-family protein. And through sequence analysis, they also showed that the pyrin domain of Caspy2 exhibited significant homology not only to pyrin domains but also to the CARDs of *Xenopus* caspase-1 (49% similarity) and bovine caspase-13 (45% similarity, previously known as human caspase-13). Thus, accordingly, we revised the figure, and only labeled the N-terminal pyrin domain of caspy2 (5-84 aa) according to Ref #10, and kept the label of the conserved catalytic domain (296 aa) for caspy2 in our manuscript.

At line 85 the authors jump from having caspase substrate cleavage data to seemingly assuming that this arises from caspy2 and not other caspases. They state that caspy2 is 57% similar to human caspase-5, but do not state whether this is significantly higher than its similarity to other human caspases, and also do not state whether caspy and caspase-C might also be similar to human caspase-5. So the rationale for focusing on caspy2 was not described, and seems like an unjustified leap in logic.

Reply: Thank you for the constructive suggestions. We have revised them accordingly (page 6, line 97-102).

Authors refer to EIB202 strain in the figure panels and in the text and never define this as "wild type" except in the methods section. This makes for a confusing read. This strain could simply be referred to as "wild type". Actually, the authors do sometimes refer to the strain as wild type in other areas of the paper.

Reply: We apologize for the confusion. To more consistent with the mutant strain 09091 in our figure panels, we used the EIB202 as wild-type throughout our manuscript. Accordingly, we have revised the text and defined this wild-type strain as EIB202 as in figure panels, to distinct from the wild-type and KO cell lines used in our study.

The extent of results text on fig. 2c is: "Moreover, 09091 *E. piscicida* infection induced caspy2 expression and the cleavage of pro-caspy2, as determined by immunoblotting using an antibody that recognizes p20, the cleaved form of caspy2 (Fig. 2c)." The authors fail to distinguish between transcriptional induction of caspy2 protein expression, as compared to activation of the protein measured by p20 protein band processing. They also do not mention that WT bacteria are also on the blot. The blot shows that transcriptional induction is driven by both WT and mutant bacteria, but processing only by the mutants. This actually bolsters the authors case. This oversight and several others give the impression that the authors have not put a lot of time into crafting the text of this manuscript.

Reply: Thank you for the constructive suggestions. We have revised them accordingly. (page 7, line 122-128).

"Taken together, these findings indicated that the existence of pyroptosis in zebrafish is mediated by caspy2 activation during bacterial infection." Given that the authors show that WT bacteria do not cause pyroptosis, this sentence is actually incorrect.

Reply: We have referred to the 09091 *E. piscicida* in the revised manuscript.

Human caspase-4/5 and mouse caspase-11 have a CARD... "However, the N-terminal domain of caspy2 is more homologous to the pyrin death domain (ref 10) (Supplementary Fig. 3)." Supp Fig. 3 does not show what this sentence says it shows. That information is in reference 10, but Supp Fig. 3 only aligns caspy2 to caspases, not to any pyrin domains.

Reply: Per the reviewer's suggestion, we have removed the citation of Supplementary Fig. 3, as Ref #10 (Masumoto et al., J. Biol. Chem. 2003, 278 (6): 4268-4278) have already revealed N-terminal of caspy2 exhibited significant homology to pyrin domains.

Fig. 4d and 4e are called figure 3d and 3e in the text. Please check figure callouts.

Reply: We have checked and revised all the figure callouts accordingly.

Fig. 6a and 6b contradict each other. 6a shows that WT bacteria do not induce expression of caspy2, while mutant bacteria do induce expression. Yet 6b shows that total caspy2 protein levels are actually somewhat higher in the WT bacteria. This data need to be clarified or removed from the paper. I think the western blot is more definitive and important as it shows caspy2 processing, and it is consistent with earlier data in Fig. 2c where both WT and mutant bacteria induce expression of caspy2 pro-protein. That said, exclusion of dead fish (see below) could be problematic in 6b as well.

Reply: We agree with the reviewer's point, and we have removed the panel accordingly.

Figure 6D. The authors have clarified in the figure legend that dead fish were excluded. However, these still need to be shown on the graph - the reader needs to be able to see how many dead fish were in the experiment. So, assuming that at the 72 hour time point 60% of the fish were dead, there should be something like 6 symbols of "X" shown together at the top of the graph (perhaps above the top of the Y axis) to represent dead and uncountable fish, and there should be 4 symbols of "O" showing the counts of the remaining live fish. However, I have the impression that the authors did not keep track of how many dead fish there were in the experiment, and that these were simply discarded without counting. I have this impression because at 72 hours it seems that there are equal numbers of dots between the Caspy2 MO and the Control MO. However, 60% of the former should be dead whereas only 10% of the Control MO should be dead. This suggests that the authors increased their sampling of the Caspy2 MO fish to obtain more live fish. This is not appropriate because it includes selection bias against the dead fish. Such a method would invalidate the time points where numbers of dead fish were very different from each other, so the 48 and 72 hour time points are unpublishable. If the authors have the numbers of dead fish from these experiments, and can include them on the graph, that is acceptable. If such data is not available, then the 48 and 72

hour time points should be removed from Figure 6D. This also affects Figure 6A and 6E and thus the 48h and 72h time points should be removed. None of this significantly affect the authors conclusions, it is simply not scientifically correct to present data from time points where dead animals are excluded.

Reply: We agree with the reviewer's point, and we have removed the data of 48h and 72h time points accordingly, since none of this significantly affect the authors conclusions.

Please remove Supp Figure 10a-10c because the data needs to be further investigated before it is published. By comparing Fig. 6c we can see that Caspy2 MO fish only 40% survive when infected with 10^5 CFU. One would expect that if the dose were increased, more fish should die. Yet at the 10^9 dose in Supp Fig. 10a-c, fewer fish die, with about 65% surviving. This makes no sense. If one does not compare Fig. 6c to Supp 10a, then each figure alone makes sense: 6c shows us that caspy2 promotes resistance to infection; 10a shows us that caspy2 is detrimental in sepsis. However, together they make no sense as a 10,000 fold increase in the dose should not cause more fish to survive. We must hypothesize that the obscenely high dose caused a secondary defense pathway to be activated, and this pathway exerted a dominant effect over the defect in caspy2 and that this secondary pathway rescued the fish from death. That needs to be worked out before the authors publish Supp Fig. 10a-c. That can be done in future work, and thus I recommend removing Supp Fig. 10a-c from this paper.

Reply: We agree with the reviewer's point, and we have removed the data for Supplementary Fig. 10a-c accordingly.

To me, the authors do not need to identify the gasdermin effector of pyroptosis that is downstream of caspy2 in this paper. I think the point that caspy2 is the LPS sensor can be made without that knowledge. That could be done in a subsequent study.

Reply: We appreciate all your effects on our manuscript.

Reviewer #2 (Remarks to the Author):

The manuscript has clearly improved a lot and I congratulate the authors on the nice LPS interaction studies. However several of my original point have not yet been fully addressed.

Point 2)

What do the authors mean by:

"In our previous data, we only primed the CTB+LPS group with LPS, because we cannot use LPS to prime and then stimulate with LPS because this might induce rapid cell death. That is why caspy2

expression was only observed when either LPS or CTB plus LPS were used.” What rapid cell death is this? LPS alone should not induce cell death, or does it? And if yes, via Caspy-2?

Reply: The reviewer raised a valid question. In our previous data, we do detect a rapid cell death, which were more similar to that apoptosis cell morphology (unpublished). Since caspy2 is a cytosolic LPS sensor, without cytosolic delivery of LPS, we expect this was result from the toll-like receptor signaling cascades. Although we feel that understanding of the pathway involved is not required for the conclusions of this manuscript and would be the subject of future investigation. Moreover, in our revised manuscript, we have performed additional experiments by priming cells with Pam3CSK4, and stimulate with LPS, CTB, or CTB+LPS, which was adopted from previous cytosolic LPS delivery study (Hargar et al. Science, 2013. 341, 1250-3), and our results do indicate that caspy2 is essential for intracellular LPS-triggered non-canonical inflammasome activation in zebrafish.

Point 3) appearance of a p20 band of Caspy-2 after infection (Figure 2c)

I believe that the catalytic site is necessary for Caspy-2 activity, but this has nothing to do with my question.

My question is: What is the p20 band, and is generation of the p20 necessary for activity.

Reply: The reviewer raised a valid question. In our manuscript, we did observe that the processing of caspy2 during non-canonical inflammasome activation (Fig 2). However, when we mutant the catalytic site, we cannot detect any cytotoxicity and processing of caspy2 as observed in Fig 4. Thus, according to our results, we expect that the processing of caspy2 might play an important role for non-canonical inflammasome activation.

Point 4) Components of the non-canonical inflammasome in Zebrafish:

To justify publication in a high impact journal like Nature Communication, I believe it is not sufficient to only show that Caspy2 binds LPS i.e. is the Zebrafish equivalent of human Caspase-4/-5, mouse Caspase-11, but the whole non-canonical pathway needs to be analyzed in detail. After all, the first papers to show that showed that LPS activates human caspase-4/-5 were published 'only' in European Journal of Immunology (Schmid-Burgk JL EJI 2015 and Baker PJ EJI 2015) despite the higher relevance of the human system.

I understand that it might be difficult to identify the Zebrafish Nlrp3 homolog, given the large number of NLRs in the fish, but I would encourage the authors to at least check 1) if zASC specks are formed after caspy-2 activation, if 2) Caspy-1 is processed after LPS transfection and if 3) this is ASC dependent, 4) if mature zIL-1b is released after caspy-2 activation and 5) if Caspy-2 processes zIL-1b directly or via Caspy-1.

These experiments should highlight similarities and differences between the human/mouse and zebrafish system, and are relevant for the in vivo infection as well as the septic shock models, since currently it is not clear which pathways downstream of caspy-2 drive the response.

Reply: Thank you for your constructive suggestions. In our manuscript, we identified the zebrafish caspy2 activation-mediated pyroptosis, both in response to cytosolic LPS and 0909I *E. piscicida* infection, but whether this caspy2-mediated non-canonical inflammasome activation can lead to “NLRP3 inflammasome” activation as observed in mammalian cells remain unknown. In adult zebrafish primary leukocytes, both caspy and caspy2 can cleave zIL-1 β in response to bacterial infection (Vojtech et al., *Infect. Immun.* 2012, 80: 2878–2885) . Moreover, zebrafish zASC interacts with caspy but not with caspy2, and the oligomerized form of the conserved adaptor protein ASC had been shown to recruit and activate caspy, the zebrafish functional homolog of caspase-1, forming a characteristic speck structure both in vitro (Masumoto et al., *J. Biol. Chem.* 2003, 278 (6): 4268-4278) and in vivo (Kuri et al., *J. Cell Biol.* 2017, 216(9): 2891-2909) . Thus, additional works still need to clarify whether the caspy2 mediated non-canonical inflammasome activation could trigger caspy processing, is this ASC dependent? And what’s the mechanism about the zebrafish IL-1 β cleavage and secretion, is this dependent on caspy2 or caspy? We do believe there must some difference between zebrafish and human/mouse system and we have discussion this in our manuscript, thus, we have added the comments above to the revised Discussion. Although this is important, we feel that understanding of the mechanism involved is not required for the conclusions of the manuscript, which should be the subject of future investigations.

Point 5) interaction of LPS-Caspy2-PYD

The authors provide very nice proof of interaction of LPS with the PYD. A last experiment would be to replace the Casp-11 CARD with the PYD from Caspy2. Would this hybrid caspase be able to sense LPS and activate?

Reply: We respectively disagree with the reviewer’s comment. Feng Shao’s work have already shown that caspase-11 CARD domain can interact with LPS (Shi et al., *Nature* 2014, 514, 187–192), and in our manuscript, we focused on revealing that the zebrafish N-terminal pyrin domain of caspy2 also plays critical role in interaction with LPS, but not Pam3CSK4. Also, we have a good control with N-terminal domain of caspy, thus, we feel the point that caspy2 is the LPS sensor can be made without that knowledge and we believe that our study has quite important implications for the death domain family, which expanding our scope of the possible functions of the domains.

Point 6) Oligomerization of Caspy2.

I find the data about oligomerization of Caspy2 more convincing than before. However, they still do not address the question whether this oligomerization induces Caspy2 processing. For example, in Fig. 3d, there is a minor band (probably Caspy2 p20) in 0909I infected cell lysate. However, it is not

visible upon LPS addition and oligomerization in Fig. 3b. It might indicate that additional players are necessary to induce Caspy 2 processing. Also, I was surprised to see that this band is not visible in Glycine-treated cells, and Caspy2 oligomers run lower in this sample.

One particular point that would be interesting to check is whether zASC is involved oligomerization of Caspy-2. In the mouse/human system non-canonical inflammasome activation does not require ASC. However in Zebrafish the ASC speck work differently, in that both ASC speck formation and caspase activation are induced via the PYD of ASC. Thus it might be possible that the PYD of Zebrafish ASC participates in the oligomerization of the caspase after LPS sensing.

Reply: Thank you for your constructive suggestions. In our manuscript, we identified the zebrafish caspy2 activation-mediated pyroptosis, and found that LPS induced caspy2 oligomerization by interaction with LPS, and according to our results, we expect that the processing of caspy2 might play an important role for non-canonical inflammasome activation. As for the role of zASC in caspy2 pathway activation, previous data have already shown that zebrafish zASC interacts with caspy but not with caspy2, and the oligomerized form of the conserved adaptor protein ASC had been shown to recruit and activate caspy, the zebrafish functional homolog of caspase-1, forming a characteristic speck structure both in vitro (Masumoto et al., J. Biol. Chem. 2003, 278 (6): 4268-4278) and in vivo (Kuri et al., J. Cell Biol. 2017, 216(9): 2891-2909) . Thus, additional works still need to clarify whether the csapy2 mediated non-canonical inflammasome activation could trigger caspy processing, is this ASC dependent? We have added the comments above to the revised Discussion. Although this is important, we feel that that understanding of the mechanism involved is not required for the conclusions of the manuscript, which should be the subject of future investigations.

Fig 2b: This does not address my issue.

The fact that a caspase-1 inhibitor doesn't block cell death in ZF4 cells doesn't mean anything unless it can be shown that it block activation of Casp-1 afetr stimulation of the canonical inflammasome pathway.

Reply: Thank you for your comments. In our manuscript, we were more focused on the identification of cytosolic LPS receptor in zebrafish, when we knockout the caspy2, the cytotoxicity was significantly reduced in our model, thus, to detected the canonical inflammasome pathways activation that downstream of or together with caspy2 could be done in a subsequent study.

Fig 3b:

Since this assay might prove to be very useful to other researchers, I would suggest to also include a schematic of the assay.

Reply: Thank you for your comments, we performed the experiments by priming cells with Pam3CSK4, and stimulate with LPS, CTB, or CTB+LPS, which were adopted from previous cytosolic LPS delivery study (Hargar et al. Science, 2013. 341, 1250-3), we have cited the references according in our revised manuscript.

Suppl. Fig 4: The response doesn't not answer my question.

Why is no LPS needed to detect caspase activity in this setting?

Reply: We performed the experiments according to a previous study (Masumoto et al., J. Biol. Chem. 2003, 278 (6): 4268-4278), which was the first identification of zebrafish Caspy and Caspy2. Accordingly, the caspases enzymatic site is correlated with their caspase-type-specific substrates activity. Our results also claimed that the enzymatic activity site is critical for WEHD activity. And we believe this also provide a clue that the processing of caspy2 might play an important role for caspy2 activation. Although this is important, we feel that that understanding of the mechanism involved is not required for the conclusions of the manuscript, which should be the subject of future investigations.

REVIEWERS' COMMENTS:

Reviewer #1 (Remarks to the Author):

The authors have addressed all my prior concerns. I think this is a very nice work, and an important advance for the field. It adds direct sensing by PYD domains to the canon of inflammasome signaling. I think this is very appropriate for Nature Communications, and congratulate the authors.

Reviewer #2 (Remarks to the Author):

I have no further comments.